# Modeling the *ACVR1*[R206H] mutation in human skeletal muscle stem cells

Emilie Barruet[1,2,3]*, Steven M Garcia[1], Jake Wu[1], Blanca M Morales[2], Stanley Tamaki[1], Tania Moody[2], Jason H Pomerantz[1,3], Edward C Hsiao[2]*

[1]Division of Plastic and Reconstructive Surgery, Departments of Surgery and Orofacial Sciences, the Program in Craniofacial Biology, and the Eli and Edythe Broad Center of Regeneration Medicine, University of California - San Francisco, San Francisco, United States; [2]Division of Endocrinology and Metabolism, Department of Medicine, the Institute for Human Genetics, the Program in Craniofacial Biology, and the Eli and Edythe Broad Center of Regeneration Medicine, University of California - San Francisco, San Francisco, United States; [3]Division of Plastic and Reconstructive Surgery, Department of Surgery, University of California - San Francisco, San Francisco, United States

**Abstract** Abnormalities in skeletal muscle repair can lead to poor function and complications such as scarring or heterotopic ossification (HO). Here, we use fibrodysplasia ossificans progressiva (FOP), a disease of progressive HO caused by *ACVR1*[R206H] (Activin receptor type-1 receptor) mutation, to elucidate how ACVR1 affects skeletal muscle repair. Rare and unique primary FOP human muscle stem cells (Hu-MuSCs) isolated from cadaveric skeletal muscle demonstrated increased extracellular matric (ECM) marker expression, showed skeletal muscle-specific impaired engraftment and regeneration ability. Human induced pluripotent stem cell (iPSC)-derived muscle stem/progenitor cells (iMPCs) single-cell transcriptome analyses from FOP also revealed unusually increased ECM and osteogenic marker expression compared to control iMPCs. These results show that iMPCs can recapitulate many aspects of Hu-MuSCs for detailed in vitro study; that ACVR1 is a key regulator of Hu-MuSC function and skeletal muscle repair; and that ACVR1 activation in iMPCs or Hu-MuSCs may contribute to HO by changing the local tissue environment.

**\*For correspondence:**
emilie.barruet@ucsf.edu (EB);
Edward.Hsiao@ucsf.edu (ECH)

## Editor's evaluation

The manuscript by Barruet et al., investigates an interesting and rare skeletal muscle dystrophy (FOP). They use both primary and induced (iPSC) muscle stem cells to determine how regeneration/engraftment is affected in this condition. The authors use a blend of histology and transcriptional approaches to determine ECM remodeling and myogenic capacity from both cell/tissue lines. The experiments are well conducted and use strong approaches and statistical measures to test their hypothesis. Overall, this is a quality manuscript on a rare muscle disease that will establish a novel model to study FOP and initial data to elucidate the molecular pathology of HO.

## Introduction

Human diseases of skeletal muscle are major medical problems. Aberrant repair after muscle injury can results in scarring, fibrosis, lipid infiltration, or heterotopic ossification (HO; bone formation in an inappropriate site). Although HO is a relatively uncommon complication, it can form after trauma, burns, spinal cord injuries, and surgical procedures such as hip arthroplasty (*Agarwal et al., 2016*; *Matsuo et al., 2019*; *Meyers et al., 2019*; *Sullivan et al., 2013*). Recently, HO has been reported in

patients with severe COVID-19, suggesting a critical role for inflammation and muscle damage (*Aziz et al., 2021*).

To better understand the link between skeletal muscle and HO, we studied fibrodysplasia ossificans progressiva (FOP) as a prototypical condition. FOP is a congenital disease of abnormal skeletal muscle regeneration and severe HO, in which the majority of patients have the classical *ACVR1*$^{R206H}$ (c.617>A) mutation (*Shore Eileen et al., 2006*) that causes hyperactivation of the BMP-SMAD signaling (*Billings et al., 2008*) and aberrant responses to Activin A (*Hatsell et al., 2015*). While previous studies suggested that multiple cell types including endothelial cells (*Barruet et al., 2016*; *Medici et al., 2010*), mesenchymal progenitors (*Culbert et al., 2014*; *Dey et al., 2016*), pericytes (*Cai et al., 2015*), and more recently mast cells (*Brennan et al., 2018*; *Convente et al., 2018*), macrophages (*Barruet et al., 2018*; *Matsuo et al., 2021*), and FAPs (*Lees-Shepard et al., 2018*) can indirectly or directly contribute to HO in FOP, how the *ACVR1*$^{R206H}$ mutation impairs human muscle repair and muscle stem cell function has yet to be determined.

Disrupted signaling of bone morphogenetic proteins (BMPs), originally identified by their ability to induce bone formation when injected into muscle (*Urist, 1965*), changes muscle homeostasis (*Ono et al., 2011*) by controlling the proliferation and differentiation of satellite cells (SCs) (*Ono et al., 2011*; *Stantzou et al., 2017*). SCs marked by PAX7 (Paired Box 7) (*Seale et al., 2000*), are a prerequisite for skeletal muscle regeneration (*Mauro, 1961*) and are thought to be the main human muscle stem cells (Hu-MuSCs) of postnatal skeletal muscle. Upon injury, activated SCs give rise to myoblasts, which form new myofibers or fuse to existing muscle fibers to repair muscle damage (*Kuang et al., 2007*). A subset of SCs does not differentiate and serves to replenish the SC pool. The BMP pathway is critical for maintaining *PAX7* expression in primary MuSCs and for preventing commitment to myogenic differentiation (*Friedrichs et al., 2011*). Abrogation of BMP signaling in SCs slowed myofiber growth (*Stantzou et al., 2017*), and increased BMP4 levels in Duchenne's muscular dystrophy (DMD) can exacerbate the disease (*Shi et al., 2013*). However, how the BMP pathway regulates muscle repair or SC function in conditions of HO such as FOP remains unclear and is a major knowledge gap.

Since primary tissues can only and safely be obtained from deceased FOP patients because of the risks of heterotopic ossification at the surgical site, patient-derived human-induced pluripotent stem cells (hiPSCs) provide a strategy to create FOP cells for in vitro and mouse xeno-transplant studies. Interestingly, BMP signaling is a key pathway that can be manipulated to make Hu-MuSCs-like cells from hiPSCs (iMPCs) (*Chal et al., 2016*; *Chal et al., 2018*; *Xi et al., 2017*). Recent protocols to create Hu-MuSCs-like cells from hiPSCs (*Takahashi et al., 2007*) could generate skeletal myogenic lineage cells, but with limited muscle regenerative capacity (*Borchin et al., 2013*; *Magli et al., 2017*; *Shelton et al., 2014*; *van der Wal et al., 2018*; *Xi et al., 2017*). PAX7 cell engraftment capability was often not reported or showed limited success. Transgene-induced *PAX7* or *PAX3* expression in hiPSCs can increase engraftment of PAX7$^+$ cells and contribution to the SC pool (*Al Tanoury et al., 2020*; *Wu et al., 2018*); however, these engineered cells may not reflect physiology due to the genetic manipulation of these master transcription factors.

In this study, we used primary human muscle stem cells isolated from deceased FOP subjects and hiPSC-derived muscle progenitor cells from FOP subjects to elucidate how *ACVR1*$^{R206H}$ affects muscle stem cell function and regenerative ability. Primary FOP Hu-MuSCs can engraft and regenerate injured muscle of immunocompromised (NSG) mice, but at lower efficiency than Hu-MuSCs from non-FOP donors, and the source of FOP Hu-MuSCs (biceps vs. diaphragm) appears to impact their engraftment efficiency. A new non-transgenic strategy for creating human iMPCs was developed and applied to multiple hiPSC lines from control subjects and subjects with FOP. These iMPCs shared transcriptional similarities with primary Hu-MuSCs, and abnormal activation of the BMP pathway by *ACVR1*$^{R206H}$ changed the transcription profiles of the FOP iMPCs compared to controls.

## Results

### Lower efficiency engraftment of primary human FOP Hu-MuSCs

Muscle tissue samples were obtained from FOP cadavers which allowed the study of markers in situ and the isolation of primary human cells with endogenous activated ACVR1 signaling. Hematoxylin and eosin and alcian blue staining of FOP primary muscle samples from two deceased FOP subjects in a region without HO showed no gross defects. FOP muscle tissue near a HO lesion showed increased

ECM proteoglycan components (alcian blue staining) in the interstitial space of the muscle fibers near the HO lesion (*Figure 1A*). Primary Hu-MuSCs carrying the *ACVR1^R206H* activating mutation were isolated from unfixed muscle tissue from two FOP autopsies. Biceps brachii muscle, commonly affected by HO, and diaphragm muscle, one of the rare skeletal muscle sites spared from HO in patients with FOP, were analyzed. PAX7 staining confirmed the presence of SCs in human FOP biceps (*Figure 1B*, middle and right). Interestingly, a small subregion showed Collagen Type 1 expression with embedded satellite cells (*Figure 1B*, right) suggesting possible early HO formation despite absence of gross HO.

Sufficient Hu-MuSCs (*Figure 1—figure supplement 1A*) were sorted for transplant and gene expression analysis. FOP Hu-MuSCs showed lower *PAX7* expression compared to control Hu-MuSCs. *COL1A1* and *ID3* were increased in affected FOP muscle (biceps) (*Figure 1C*). However, *ID1* was increased in both the FOP diaphragm and biceps compared to control muscles (*Figure 1C*). Hu-MuSCs were transplanted into the irradiated tibialis anterior (TA) muscle of immunocompromised (NSG) mice (to hinder endogenous satellite cells) previously injured with bupivacaine (*Garcia et al., 2017*). The bupivicaine step induces myofiber injury and promotes engraftment of donor Hu-MuSCs. Five weeks after transplantation, biceps and diaphragm FOP Hu-MuSCs had engrafted and formed human fibers (*Figure 1D and E*). However, the number of human DYSTROPHIN-positive fibers was significantly lower with FOP Hu-MuSCs vs. control Hu-MuSCs (*Figure 1F*). The number of engrafted PAX7⁺ cells was qualitatively lower with FOP vs. control Hu-MuSCs (*Figure 1F* and *Figure 1—figure supplement 1B*). Ten weeks after re-injury with bupivacaine at week 5 (*Figure 1D*), the number of human DYSTROPHIN fibers was significantly decreased when FOP biceps Hu-MuSCs were transplanted compared to control Hu-MuSCs, but not when FOP diaphragm Hu-MuSCs were transplanted (*Figure 1G and H*). No differences in the number of human PAX7 cells were identified (*Figure 1G and H*). No radiologic evidence of HO was found in any mice (*Figure 1—figure supplement 1C*).

Thus, primary Hu-MuSCs isolated from two FOP subjects can engraft and regenerate injured muscle of NSG mice. Our limited sample size suggests this may occur at lower efficiency than control MuSCs. In addition, the results suggest that the source of FOP Hu-MuSCs (biceps or diaphragm) may impact MuSC engraftment efficiency.

## Human FOP iPSCs can differentiate into skeletal muscle cells

The rarity of FOP disease and difficulty obtaining human tissue samples from patients with FOP makes it difficult to obtain a reliable source of muscle stem cells. Therefore, we used established and fully characterized control hiPSCs (Wtc11, 1323–2, and BJ2) and *ACVR1^R206H* hiPSCs (F1-1, F2-3, F3-2) lines previously derived from patients with FOP (*Matsumoto et al., 2013*). We first asked whether our lines were able to differentiate into myogenic cells and specifically into muscle stem cell-like cells using an adapted version of recently published protocols (*Hicks et al., 2018*; *Shelton et al., 2014*; *Figure 2A*). The control hiPSCs yielded PAX7, MYOGENIN-expressing cells (*Figure 2B and C* and *Figure 2—figure supplement 1A*), expressed DYSTROPHIN (*Figure 2B*), and formed contractile myotubes (*Figure 2C*; *Video 1* and *Video 2*).

Since BMPs control skeletal muscle differentiation from hiPSCs (*Chal et al., 2016*; *Xi et al., 2017*), we investigated if genetic activation of the BMP pathway via *ACVR1^R206H* could alter the myogenic differentiation of hiPSCs (F1-1, F2-3, F3-2) derived from patients with FOP (*Matsumoto et al., 2013*). All three FOP iPSC lines formed contractile myotubes with cells expressing PAX7, MYOGENIN, and DYSTROPHIN (*Figure 2B and C*, *Figure 2—figure supplement 1A*). *PAX7*, *MYF5*, *MYOD1*, and *MYOGENIN* (*Figure 2D*) gene expression showed some heterogeneity among the different hiPSC lines but these differences were not statistically significant. The heterogeneity seen is likely due to the presence of other cell types in our differentiation such as neuronal cells (*Figure 2—figure supplement 1C*). Adding a SMAD inhibitor of the BMP pathway (LDN193189) into the differentiation protocol (*Figure 2E*) improved the differentiation of the control hiPSC lines. LDN also improved differentiation of the FOP F3-2 (*Figure 2F*) line, which had shown lower *PAX7* and *MYOGENIN* expression (*Figure 2D*). These findings show that individual hiPSC lines are heterogeneous in differentiation to skeletal muscle lineages, similar to other protocols (*Volpato and Webber, 2020*), that FOP hiPSCs can form skeletal muscle cells and Hu-MuSC-like cells expressing PAX7 despite upregulation of the BMP pathway by *ACVR1^R206H*, and that chemical blockade of the BMP pathway can improve the formation of Hu-MuSC-like cells from the FOP iPSC line that showed the lowest efficiency.

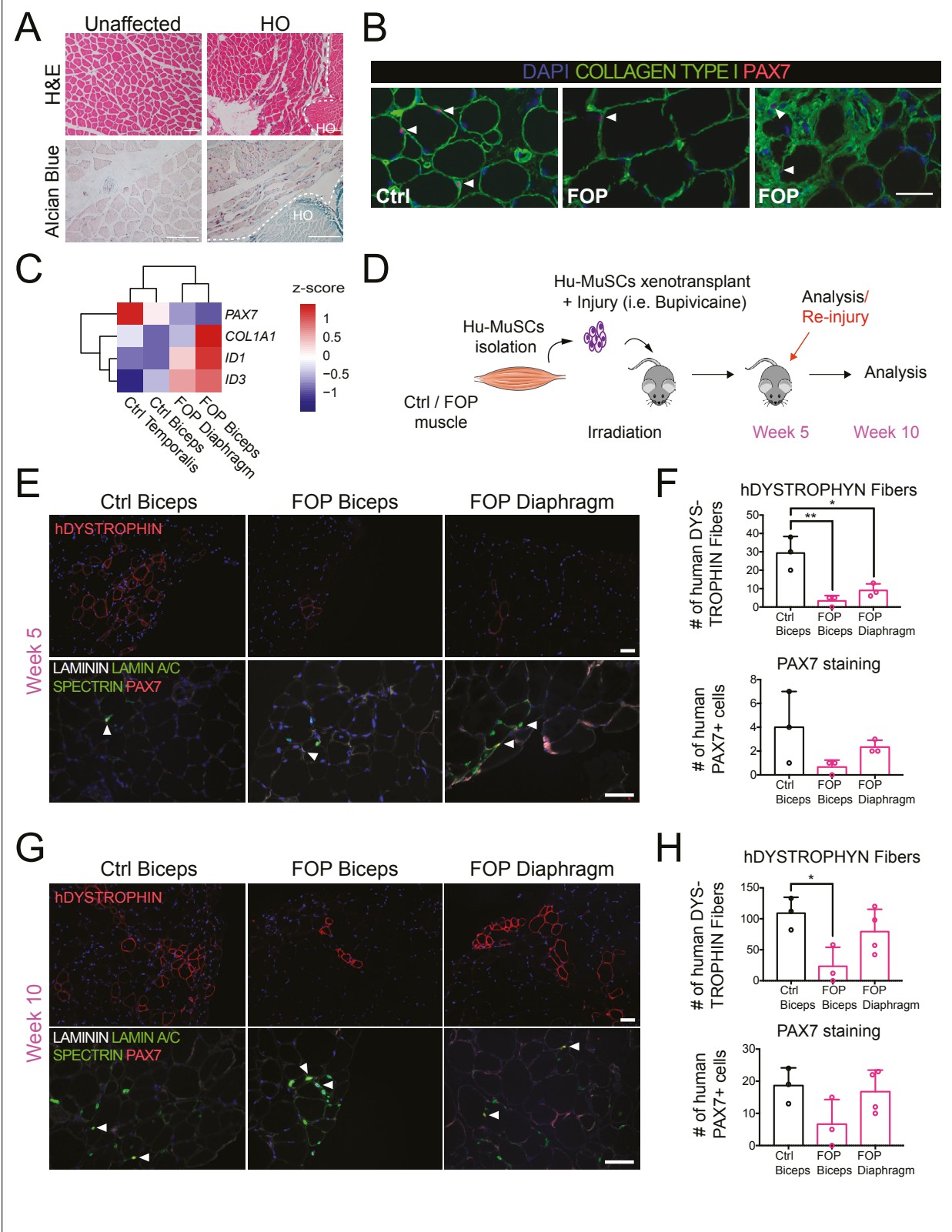

**Figure 1.** Regenerative properties of primary Hu-MuSCs isolated from FOP patients. (**A**) Hematoxylin and eosin staining (top) and alcian blue staining (bottom) of muscle cross sections from FOP subjects, with no heterotopic ossification (left) and with heterotopic ossification, depicted by the white dashed line (right, 100 µm scale bar). (**B**) Immunofluorescence staining for PAX7 and COL1 of muscle cross sections from control and FOP subjects. White arrowheads mark Hu-MuSCs, 50 µm scale bar. (**C**) Heat map of normalized gene expression of sorted human satellite cells from two control

*Figure 1 continued on next page*

*Figure 1 continued*

subjects and two different muscles of 1 FOP subject. (**D**) Schematic and experimental time course. Hu-MuSCs from biceps and diaphragm muscles of a deceased FOP patient were sorted and transplanted into NSG mice. (**E**) Immunofluorescence staining 5 weeks after xenotransplantation. White arrowheads mark Hu-MuSCs, 50 µm scale bar. (**F**) Quantification of human DYSTROPHIN fibers and human PAX7+ cells 5 weeks after transplant. (**G**) Mice were re-injured 5 weeks after transplantation with bupivacaine. Immunofluorescence staining was performed at week 10. White arrows mark Hu-MuSCs, 50 µm scale bar. (**H**) Quantification of human DYSTROPHIN fibers and human PAX7+ cells after re-injury at week 10. n = 1, biological replicates, n ≥ 3 technical replicates. Error bars represent mean and SD. *, p < 0.05, **, p < 0.01. Muscle specimen and transplantation details are listed in *Figure 1—source data 1* and *Figure 1—source data 2* .

The online version of this article includes the following source data and figure supplement(s) for figure 1:

**Source data 1.** Muscle specimen information.

**Source data 2.** Hu-MuSCs transplantation details.

**Figure supplement 1.** Transplanted Hu-MuSCs from biceps and diaphragm of FOP patient do not form bone.

## PAX7 expressing cells can be isolated from myogenic differentiation

To test the regenerative properties of PAX7-expressing MuSCs, we used FACS to purify HNK1-CD45-CD31- cells co-expressing CD29, CXCR4, and CD56 markers present on human PAX7+ cells (*Garcia Steven et al., 2018*; *Figure 3A* and *Figure 3—figure supplement 1A–C*). FACS analysis identified intermediate CD56 cells expressing high *PAX7* and low *MYOGENIN* (*Figure 3B* and *Figure 3—figure supplement 1C,D*), consistent with Hu-MuSC expression profiles. All HNK1-CD45-CD31- CXCR4+CD29+CD56dim cells formed myotubes expressing MHC (*Figure 3—figure supplement 1E*) when cultured in terminal differentiation media demonstrating isolation of functional iMPCs with satellite cell-like characteristics from the cultures.

## Isolated iMPCs can regenerate injured mouse muscle and form human fibers

To assess iMPC regenerative capacity in vivo, we injected 1000–10,000 iMPCs derived from control or FOP hiPSCs (*Figure 3—source data 1*) into the irradiated TA muscle of NSG mice. New fibers expressing human DYSTROPHIN and PAX7 cells were found after 5 weeks, showing that iMPCs could engraft and promote muscle regeneration (*Figure 3C*). However, the number of human fibers and human PAX7+ cells remained low (*Figure 3D*) compared to primary Hu-MuSCs. While some iMPC transplants yielded up to 60 new human fibers, some did not yield any human fibers. By comparison, 2000 primary non-FOP Hu-MuSCs resulted in an average of 155 human fibers based on prior assessments using the same assay (*Garcia Steven et al., 2018*). No significant differences between control and FOP iMPCs were identified, though some individual FOP samples showed higher engraftment (*Figure 3D*).

These results showed that iMPCs have engraftment potential into a muscle injury site in mice, but engraftment efficiency may be lower than primary Hu-MuSCs or be the result of differences in experimental conditions. Also, *ACVR1R206H* did not significantly impact muscle fiber regeneration in this assay.

## Transcriptional profiling of iMPCs

The lower engraftment of iMPCs compared to primary Hu-MuSCs suggested that the FACS-purified population was still heterogeneous or that iMPCs do not fully recapitulate adult primary Hu-MuSCs. Single cell RNA sequencing (scRNAseq) from control (1323–2) and FOP (F3-2) iMPCs (these lines were selected based on their lower intra-line variability, *Figure 2*) were analyzed (*Figure 4A* and *Figure 4—source data 1*). Cell populations for both samples were defined by the dimension reduction technique of uniform manifold approximation and projection (UMAP) (*Becht et al., 2018*) and unsupervised clustering with Seurat v3 package (*Stuart et al., 2019*; *Figure 4—figure supplement 1A–B*). Both control (*Figure 4—figure supplement 1A*) and FOP (*Figure 4—figure supplement 1B*) samples had clusters expressing myogenic genes (*PAX7* and *MYOD*); mesenchymal genes (*PDGFRA*); and neuronal genes (*SOX2*) (*Figure 4—figure supplement 1C*). Merged analysis to allow direct comparison identified 13 distinct clusters (*Figure 4B* and *Figure 4—figure supplement 1D*). Cells expressing muscle markers (*PAX7, MYF5, MYOD,* and *MYOG* [*MYOGENIN*]) were found in clusters 0–2 (*Figure 4B–C* and *Figure 4—figure supplement 1E*). Mesenchymal genes (*PDGFRA, ASPN,* and *COL1A1*) were

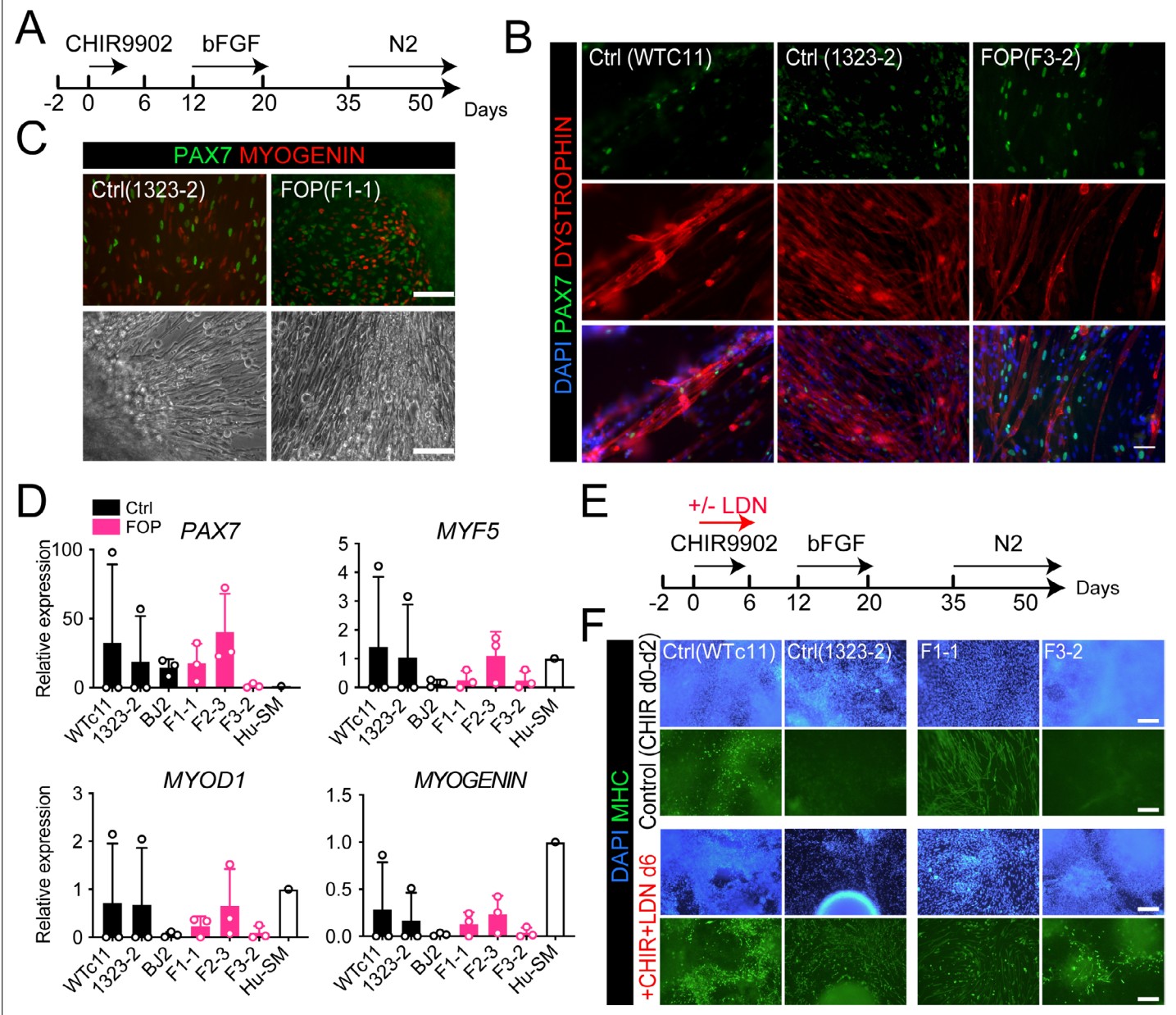

**Figure 2.** Skeletal muscle differentiation of human control and FOP hiPSC lines. (**A**) Differentiation schematic of hiPSCs into skeletal muscle stem cell-like cells (iMPCs). (**B**) Immunofluorescence staining for PAX7, DYSTROPHIN and nuclear stain DAPI of iMPCs at day 50, 50 μm scale bar. (**C**) Immunofluorescence staining of PAX7 and MYOGENIN expressing cells. Control and FOP hiPSCs can form contractile myotubes, 100 μm scale bar. (**D**) *PAX7*, *MYF5*, *MYOD1*, and *MYOGENIN* gene expression at day 50 of differentiation (n = 3 biological replicates and n ≥ 3 technical replicates). Error bars represent mean ± SD. No statistically significant differences were detected. (**E**) Schematic describing the addition of the BMP pathway inhibitor LDN193189 (LDN). (**F**) Representative immunofluorescence staining of 2 control and 2 FOP hiPSC lines differentiated and stained for MHC (Myosin Heavy Chain) and nuclear stain DAPI at day 50±LDN addition, 200 μm scale bar.

The online version of this article includes the following figure supplement(s) for figure 2:

**Figure supplement 1.** Expression of muscle and neuronal markers during the myogenic differentiation.

expressed in cluster 3. Cluster four consisted cells expressing retinal (*OTX2*) and neuroprogenitor cell (NPC) markers (*SOX2, MAP2*). Neuroprogenitor markers (*SOX2, MAP2*) and *HES6* were upregulated in cluster 5. Clusters 6 and 7 were made of cycling neuroprogenitor cells (*MKI67, TOP2A*). We also found three clusters (8-11) to be neuroepithelial cells (NECs), with cluster 10 having and increased in expression of mTor gene downstream target. Glial cells (*TNC, SLC1A2*) were comprised in cluster 12 (*Figure 4C and D* and *Figure 4—figure supplement 1E*). The frequency of muscle cells (clusters

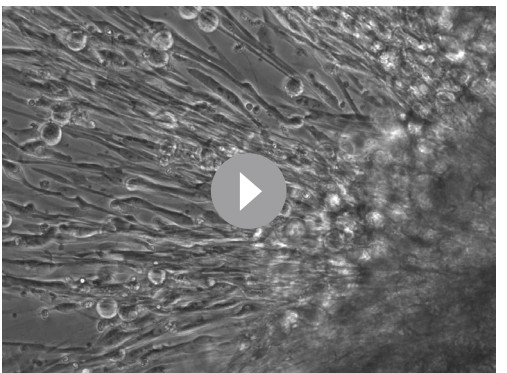

**Video 1.** Myotube contraction of differentiated Control (1323-2) hiPSCs.

https://elifesciences.org/articles/66107/figures#video1

0–2) was higher in FOP (14%, 2.5%, and 1.7%) compared to controls (4.2%, 0.5%, and 0.5%) (*Figure 4F*) suggesting that FOP hiPSCs may be more efficient at making muscle progenitor cells. Thus, FACS-purified iMPC cultures contain muscle stem/progenitor cells, but other cell types such as mesenchymal, neuronal progenitor cells, and myoblasts persist in the culture. Furthermore, the frequency of iMPCs appears to be higher in FOP vs. control cell cultures.

## FACS-sorted iMPC transcriptome is heterogeneous

Typical muscle progenitor cultures are expected to contain cells undergoing expansion, differentiation, and maturation. Sub-clustering (*Figure 5A and B*) identified five new myogenic subpopulations (*Figure 5A and B*). Clusters 0–3 expressed higher levels of *PAX7* and *MYF5* (markers of more quiescent MuSCs) while cluster four expressed higher levels of *MYOD* (or *MYOD1*), *MYOG*, *SOX8*, and *MEF2C* (markers of differentiated MuSCs/myoblasts/myocytes). Muscle stem-like cells expressing *APOE*, *KRT17*, and *CAV1* defined cluster 0. Quiescence markers (*SPRY1*, *HEY1*, *HES1*) were also highly expressed in this cluster. *CDH15*, a niche regulator of SCs quiescence (*Goel et al., 2017*) was enriched in cluster 1. Cluster 1 also had increased expression of *DES*, *CHRNA1*, and *MYOD1* suggesting that these cells have a transcriptome resembling the profile of progenitor cells. Cluster 2 consisted of muscle stem-like cells expressing high levels of *FOS* (*Figure 5B* and *Figure 5—figure supplement 1A*). Cell cycle (*TOP2A* or *KI67*) markers were increased in cluster 3 (*Figure 5B* and *Figure 5—figure supplement 1B* (left)). Cluster 3 had a higher proportion of cells in G2M and S phase (*Figure 5—figure supplement 1B* (left)). The cell cycle distribution was similar in control and FOP (*Figure 5—figure supplement 1B* (right)). The proportion of cells in clusters 3 and 4 were similar in both control (10% and 12.7%) and FOP (14.3% and 9%) samples. The proportion of cells in clusters 0 and 3 was higher in FOP (31.4% and 23.4%) compared to control (20% and 5.5%), while the proportion of cells in cluster two was increased in control (51.8% vs 21.9%) (*Figure 5—figure supplement 1D*).

We ordered the myogenic cells into three major branches using the Monocle analysis package (*Trapnell et al., 2014*) based on genes that differ between clusters and constructed pseudotime differentiation trajectories (*Figure 5D*). The pseudotime ordering of the cells (*Figure 5D*) showed that cycling MuSCs (cluster 3) were located at the tip of the tree branch A while cluster 0 comprised of the most quiescent MuSCs were mainly distributed along branch A. Cells from the *FOS* upregulated MuSCs (cluster 2) were distributed along branch B. Cluster 1 (muscle progenitor cells) was distributed in the more proximal parts of branches B and C. Distal parts of branches B and C, were notably comprised of more mature cells from cluster 4 (myoblasts) (*Figure 5E*). *PAX7* and *MYF5* were upregulated in branch A but downregulated in the tip of branches B and C (*Figure 5C and E*) confirming the correct cell ordering. *MYOD* and *MYOG* showed higher expression patterns early in branch B and late in branch C (*Figure 5E and F*).

Thus, hiPSC differentiation cultures contain subpopulations of iMPCs showing the expected spectrum of quiescence, activation, and differentiation with FOP cultures having a higher proportion of cells in the stem cell/progenitor and proliferating phases and fewer mature myoblasts.

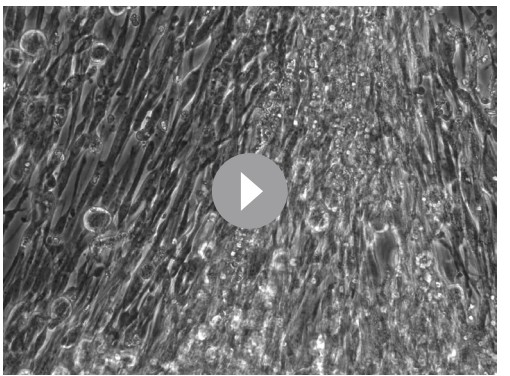

**Video 2.** Myotube contraction of differentiated FOP (F1-1) hiPSCs.

https://elifesciences.org/articles/66107/figures#video2

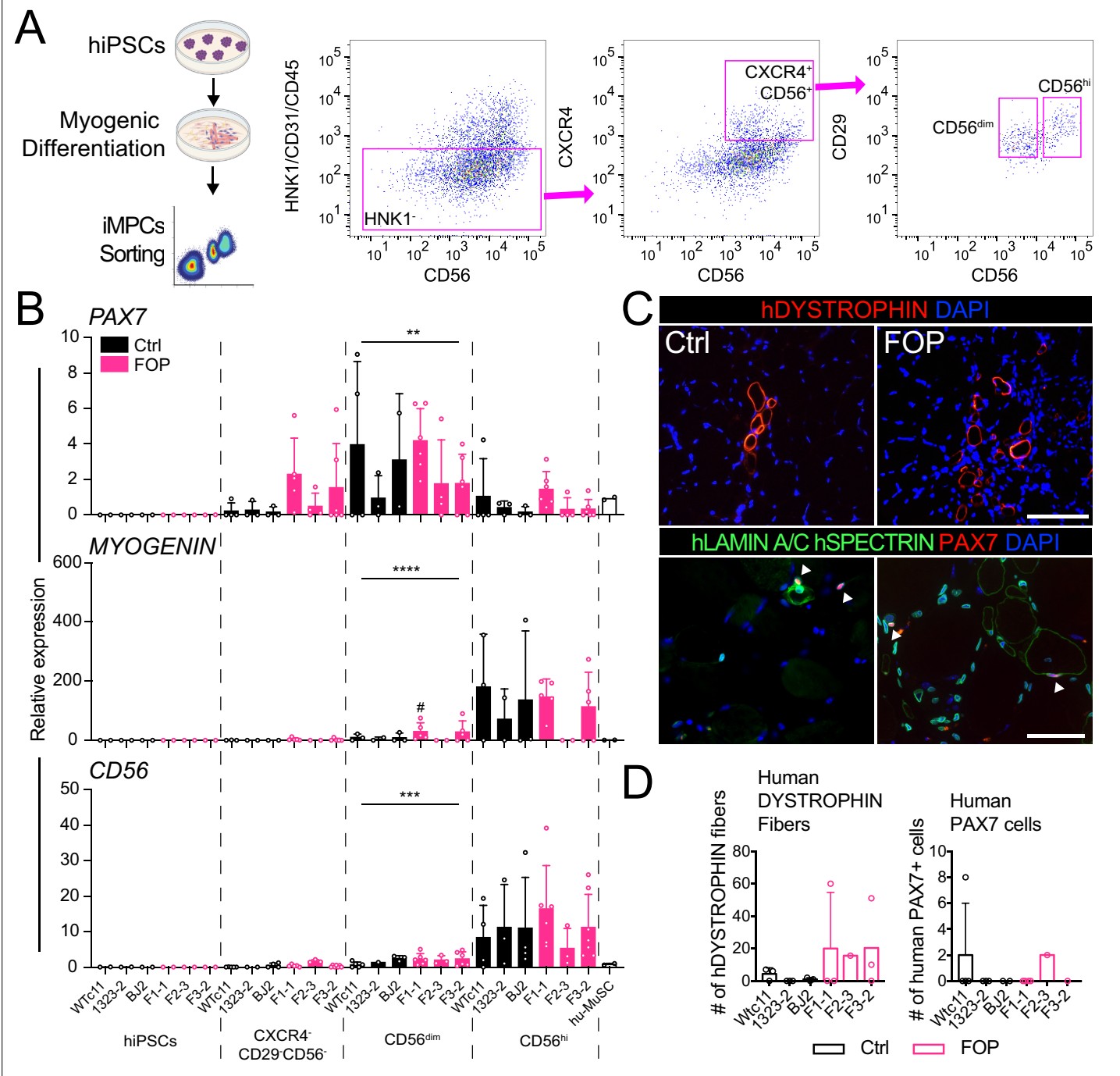

**Figure 3.** Isolation and transplantation of PAX7-expressing cells from hiPSC muscle differentiation culture. hiPSCs were differentiated into iMPCs until day 50 and sorted via flow cytometry. (**A**) Gating strategy. (**B**) Myogenic gene expression of CD56$^{dim}$ and CD56$^{hi}$ cells. CD56$^{dim}$ vs CD56$^{hi}$, n = 6 (3 Ctrl and 3 FOP lines) biological replicates, ** p < 0.01, *** p < 0.001, **** p < 0.0001. No significant differences were found between control and FOP lines. (**C**) Representative human DYSTROPHIN (top, 200 µm scale bar), and human LAMIN A/C, human SPECTRIN, and PAX7 (bottom, 100 µm scale bar) immunohistochemistry of NSG mice anterior tibialis, where sorted iMPCs were transplanted. White arrows show engrafted hiPSC-derived muscle stem cells. (**D**) Quantification of human DYSTROPHIN fibers and human PAX7 cells at week 5 after transplant (n = 3 biological replicates). Transplantation details are in *Figure 3—source data 1*.

The online version of this article includes the following figure supplement(s) for figure 3:

**Source data 1.** hiPSC-derived HNK1$^-$CD45$^-$CD31$^-$ CXCR4$^+$CD29$^+$CD56$^{dim}$ cell transplants.

**Figure supplement 1.** Differentiated cell types in the myogenic differentiation varies between cell lines, sorted cells express PAX7 and can differentiate into myotubes.

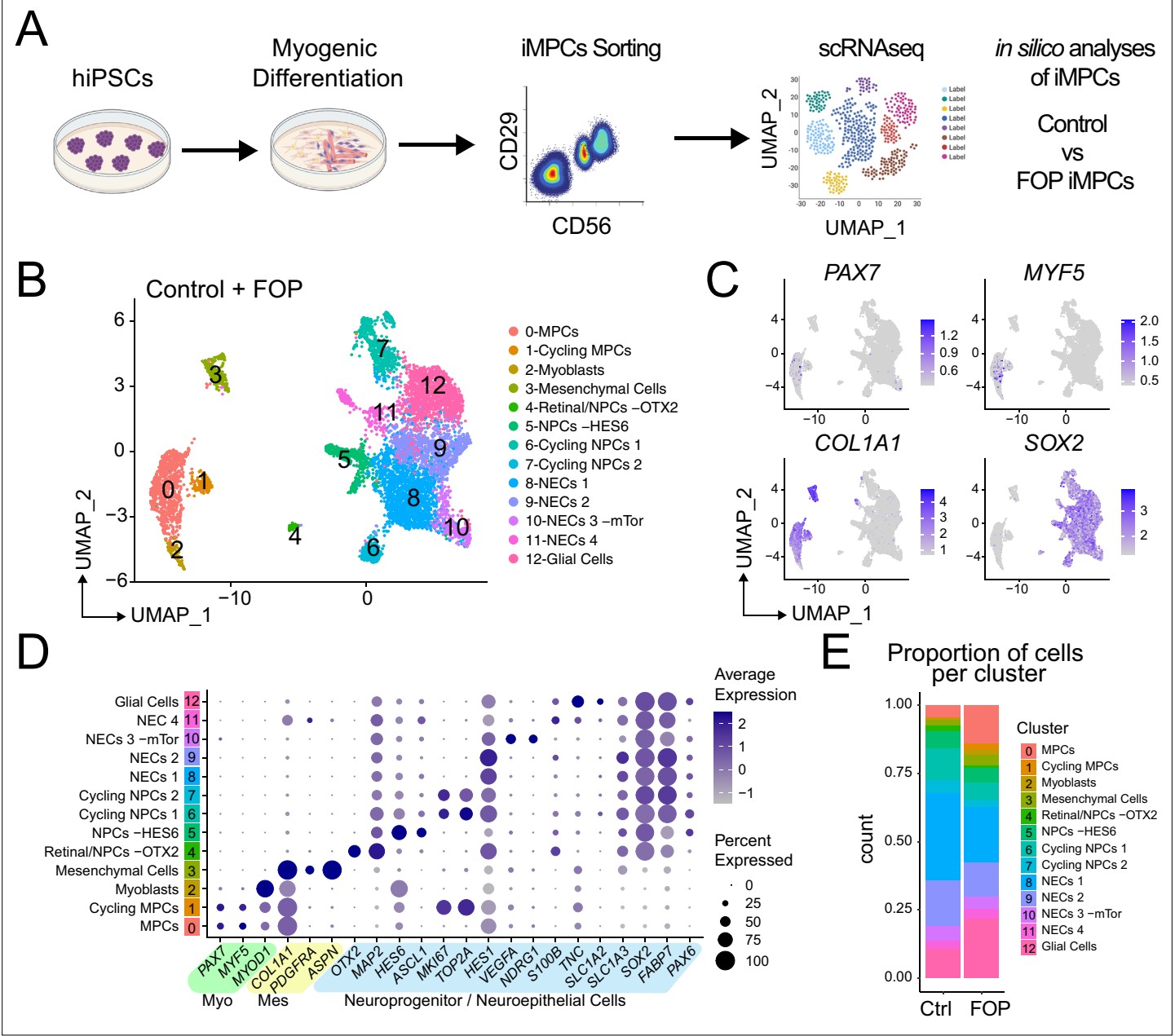

**Figure 4.** Single-cell RNA sequencing of HNK1⁻CD45⁻CD31⁻CXCR4⁺CD29⁺CD56ᵈⁱᵐ cells. (**A**) Schematic of differentiating hiPSCs, sorting of iMPCs, and scRNAseq. (**B**) UMAP visualization plots of cells combined from both control and FOP samples. Cell types were assigned based on the expression of marker genes (*Figure 4—figure supplement 1*), MPC (muscle progenitor cells), NPCs (neuroprogenitor cells) and NECs (neuroepithelial cells). (**C**) Feature expression plots showing the localization of cells expressing myogenic markers (*PAX7*, *MYF5*), mesenchymal (*COL1A1*), and neural cell marker (*SOX2*). (**D**) Dot plot displaying expression genes associated with myogenesis, mesenchymal and neurogenesis markers. (**E**) Proportion of cells per cluster for each sample.

The online version of this article includes the following figure supplement(s) for figure 4:

**Source data 1.** Quality control information for each sample.

**Figure supplement 1.** Markers expressed in sorted control and FOP HNK1⁻CD45⁻CD31⁻ CXCR4⁺CD29⁺CD56ᵈⁱᵐ cells.

## FOP iMPCs cells express increased chondro/osteogenic and ECM markers

Differential expression analysis on the transcriptional profiles of the sub-clustered myogenic cells (*Figure 5A and B* and *Figure 5—figure supplement 1A-E*) was used to see if *ACVR1^{R206H}* altered

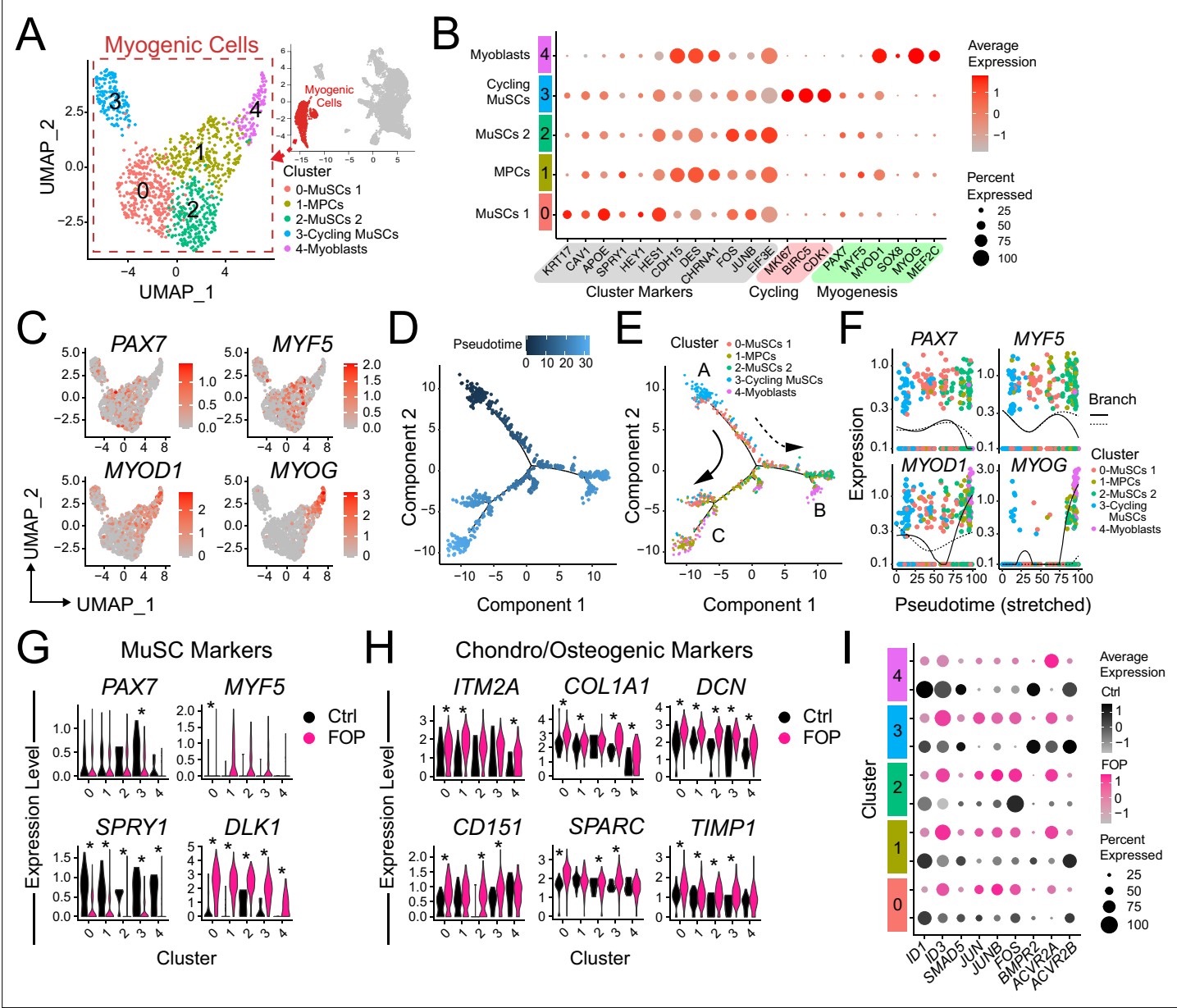

**Figure 5.** Transcriptional profile of iMPCs. (**A**) Identified myogenic clusters (in red) were sub-clustered from the other clusters (neurogenic and mesenchymal) and re-analyzed. UMAP of the new myogenic subclusters was generated. (**B**) Dot plot displaying expression of cluster defining genes. (**C**) Feature expression plots of myogenic markers. (**D**) Myogenic cells were ordered as a function of pseudotime using the Monocle package. (**E**) Pseudotime trajectory plot of myogenic clusters. Arrows represents the direction and major branches of the pseudotime. Branches A-C are marked with letters along the trajectories. (**F**) Gene plots displaying the expression of specific myogenic genes as a function of pseudotime. (**G–H**) Violin plots of myogenic (**G**) and chondro/osteogenic genes (**H**) that were significantly differentially expressed. Violin plot width depicts the larger probability density of cells expressing each particular gene at the indicated expression level. *, significantly different following differential expression testing using the Wilcoxon rank sum test per cluster. (**I**) SMAD and P38MAPK pathway markers significantly differentially expressed between control and FOP myogenic cells. p values for (**G–H**) are in *Figure 5—source data 1*.

The online version of this article includes the following figure supplement(s) for figure 5:

**Source data 1.** Differential expression analysis of control and FOP myogenic cells.

**Figure supplement 1.** Analysis of the myogenic sub-clusters.

transcriptional signatures. While *PAX7* was significantly increased in control vs. FOP cells in cluster three only, *MYF5* was significantly increased in FOP cells from cluster 1 compared to control cells (*Figure 5G*, *Figure 5—source data 1*). Since Hu-MuSCs show heterogeneous levels of PAX7 and MYF5 expression (*Kuang et al., 2007*), this suggests the *ACVR1$^{R206H}$* mutation may favor one sub-population over another. Interestingly, *SPRY1*, a known regulator of quiescence (*Shea et al., 2010*) which decreases with age (*Bigot et al., 2015*), was significantly downregulated in FOP cells in all the clusters, while *DLK1*, which act as a muscle regeneration inhibitor (*Andersen et al., 2013*) was significantly increased in FOP cells (*Figure 5G* and *Figure 5—source data 1*).

Since the *ACVR1$^{R206H}$* mutation increases BMP pathway activity and expression of chondrogenic and osteogenic markers in multiple lineages (*Barruet et al., 2016*; *Culbert et al., 2014*; *Matsumoto et al., 2013*), we examined the expression levels of extracellular matrix, fibrogenic, chondrogenic, and osteogenic genes. *ITM2A*, *COL1A1*, *DCN*, *CD151*, *SPARC*, and *TIMP1* were significantly increased in FOP cells (*Figure 5H*, *Figure 5—source data 1*). ECM proteoglycans known to be involved in inflammation, including *BGN* (*Nastase et al., 2012*) and *LUM* (*Nikitovic et al., 2014*), *TAGLN* which regulates osteogenic differentiation (*Elsafadi et al., 2016*), and *IGBP5* which is increased in aged satellite cells (*Soriano-Arroquia et al., 2016*), were also increased in FOP cells (*Figure 5—figure supplement 1E*). We also investigated chondrogenic and osteogenic genes that have been described to be increased in other FOP cell types (*Culbert et al., 2014*; *Hino et al., 2015*; *Matsumoto et al., 2013*) such as *SOX9*, *SP7*, *RUNX2*, *ACAN*, and *BGLAP*. Overall, those markers had very low level of expression in addition to have few cells expressing them. However, a higher number of cells expressed *SOX9* in FOP than control cells (*Figure 5—figure supplement 1F-G*). *SP7* was not found to be expressed in any of the samples or clusters.

Finally, expression of target genes of the BMP pathway (*ID1*, *ID3*, *BMPs*, and *SMADs*) and the p38MAPK pathway (*Figure 5I* and *Figure 5—figure supplement 1H*) was assessed to see if activated ACVR1 altered these pathways. *ID1* was significantly higher in clusters 0 (MuSCs 1), 1 (MPCs), 3 (Cycling MuSCs), and 4 (Myoblasts) in control cells, while *ID3* was significantly higher in FOP cells clusters 1 (MPCs) and 2 (MuSC 2). In addition, the BMP/TGFβ pathway downstream target gene *SMAD5* was significantly higher in cluster 2 (MuSCs 2) of control cells. The p38 pathway components *JUN* (clusters 1, 3), *JUNB* (clusters 0–2), and *FOS* (clusters 1, 3) were significantly increased in FOP cells. Within the known ACVR1 co-receptors, *BMPR2* (clusters 3, 4) and *ACVR2B* (clusters 0, 2, 3) expression were significantly higher in control cells while *ACVR2A* expression was significantly higher in FOP cells (clusters 1, 4). *ACVR2B* was significantly higher in control cells (clusters 0, 1, 4) (*Figure 5I*). Similar to the primary FOP Hu-MuSCs (*Figure 1*), these results suggest that FOP iMPCs may have a chondrogenic/osteogenic signature, increased *ID3* expression, and also showed higher p38 pathway activity and higher levels of the *ACVR2A* co-receptor at different stages of myogenic differentiation.

## iMPC transcriptome shows similarities to primary Hu-MuSCs

Comparing the iMPC scRNAseq to primary Hu-MuSCs data of sorted satellite cells from a human vastus lateralis muscle (*Barruet et al., 2020*) was used to identify if their lower engraftment efficiency was due to transcriptional differences. The merged data UMAP (*Figure 6A*) showed that clusters 0–5 contained myogenic cells (*Figure 6B and C*). *PAX7$^+$* + were identified in clusters 0–4 (*Figure 6C*). Myocyte contaminants present in the primary sorted cells constituted cluster 5. Although iMPCs expressed *PAX7* and *MYF5*, the gene expression levels were higher in primary Hu-MuSCs. In contrast, *MYOD* was higher in iMPCs (*Figure 6C* and *Figure 6—figure supplement 1*). Mesenchymal cells were identified in cluster six while NPCs, NECs and Glial cells were identified in clusters 7–9, 10–12 and 13, respectively (*Figure 6B and C*).

Detailed analysis of the myogenic cell subset (cluster 0–5, *Figure 6D*) was performed using pseudotime trajectory analysis to elucidate the states of the iMPCs with respect to primary Hu-MuSCs. Cells from clusters 0, 1, and 4 were distributed along branches B and C. Myocytes ordered at the distal end of branch A. iMPCs ordered away from primary Hu-MuSCs (*PAX7+*) and primary myocytes (*MYL1+*) (*Figure 6E–G*). Branch expression analysis modeling (BEAM) allowed us to investigate significant gene that are branch-dependent in their expression (*Figure 6—source data 1*). Branch A was mainly constituted of primary Hu-MuSCs while branch B consisted of iMPCs and subset of primary Hu-MuSCs, expressed significantly higher levels of genes associated with mesenchymal, fibrogenic, chondrogenic, osteogenic lineages and extracellular matrix (*Figure 6G and H*). Thus, iMPCs retained strong

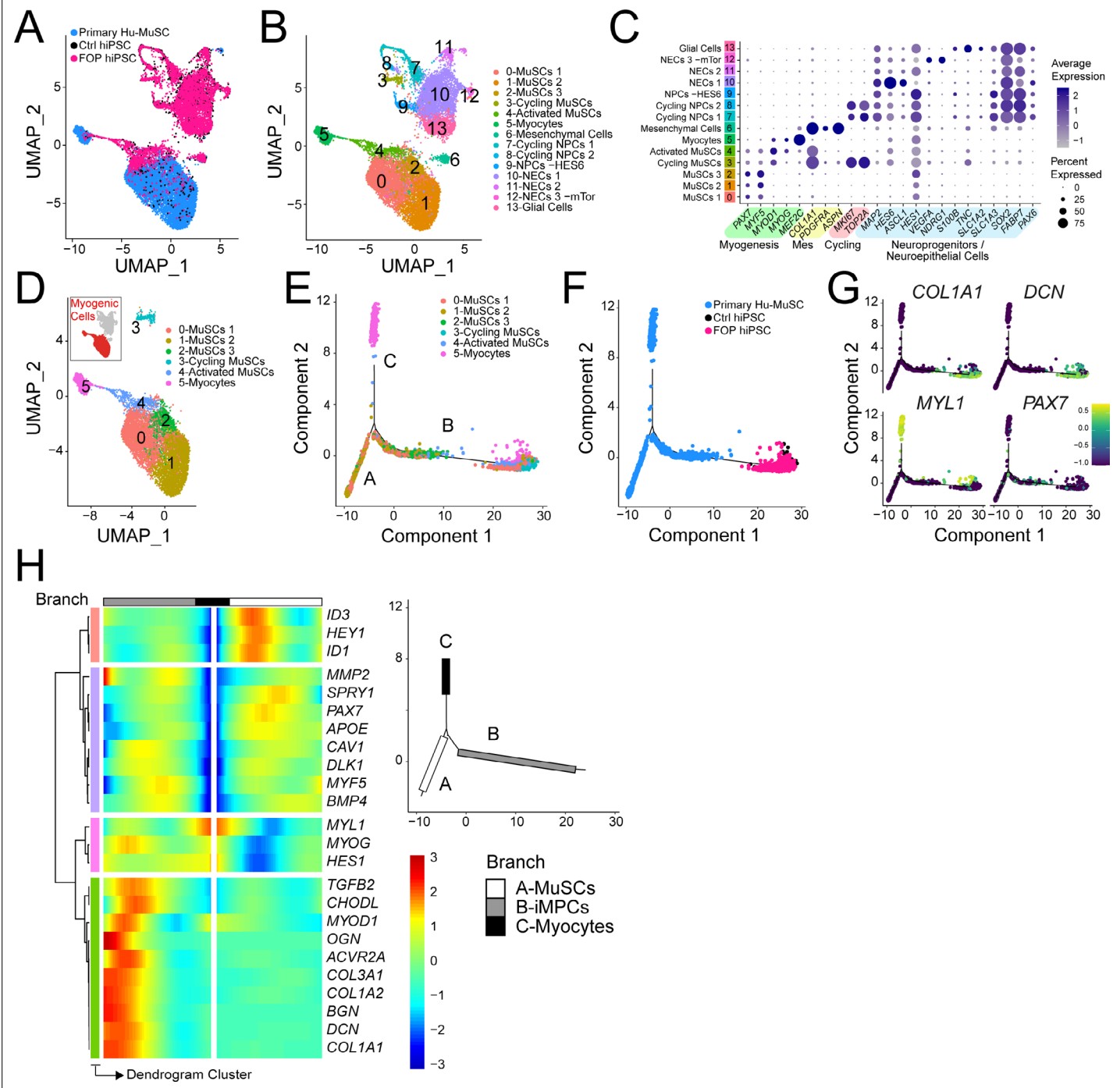

**Figure 6.** iMPC transcriptional signature compared to human primary muscle stem cells. (**A–B**) UMAP of cells combined from human primary muscle stem cells (vastus muscle) and the control and FOP samples. (**A**) UMAP showing the distribution of cells per sample, and (**B**) with clusters labeled. (**C**) Dot plot displaying expression of cluster defining genes. (**D**) Myogenic cells (in red) are comprised in cluster 0–5. UMAP of the myogenic clusters. (**E–G**) Pseudotime trajectory plot generated via Monocle analysis depicting all myogenic clusters (**E**) and samples (**F**). (**G**) Level of expression of ECM/ osteogenic genes along the cell trajectories. (**H**) Heatmap representing genes that are significantly branch dependent using the BEAM analysis (*Figure 6—source data 1*) and also genes that have similar lineage-dependent expression patterns. Branches are shown in the upper right panel.

The online version of this article includes the following figure supplement(s) for figure 6:

**Source data 1.** BEAM analysis.

**Figure supplement 1.** Expression of myogenic markers across primary and hiPS-derived myogenic cells.

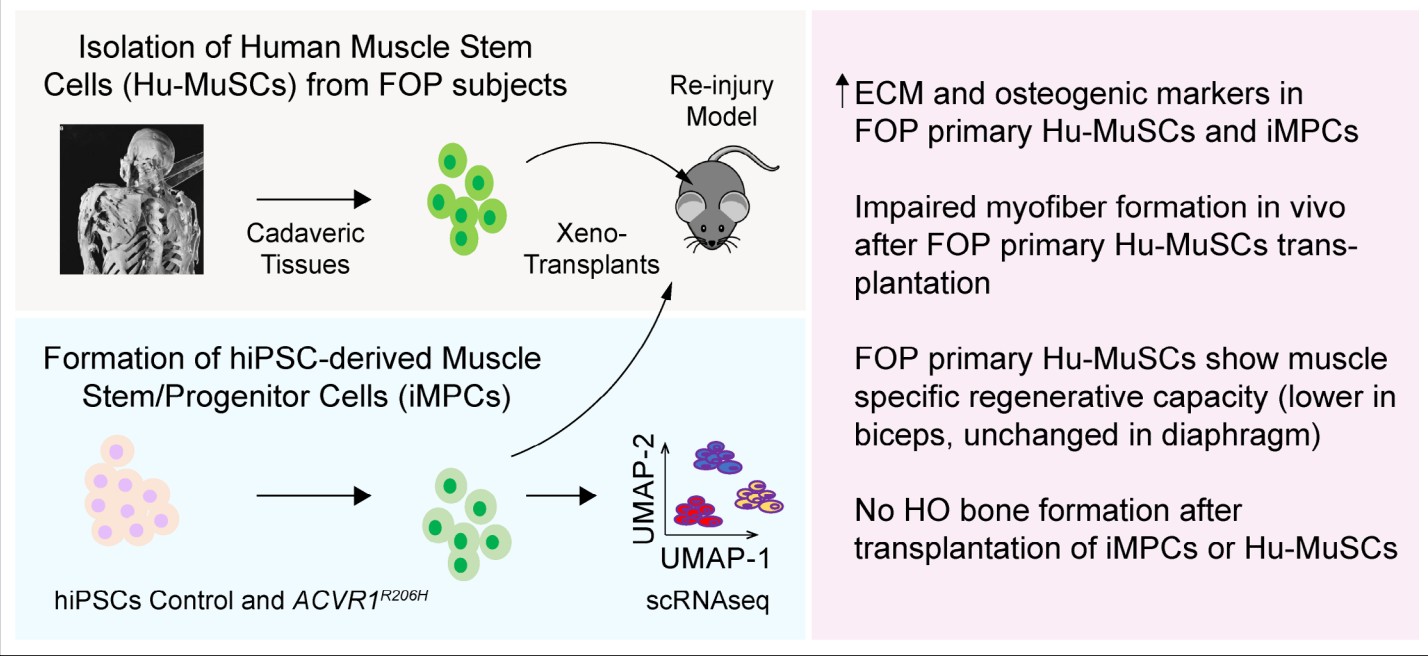

**Figure 7.** Testing the role of activated ACVR1 signaling in muscle repair. FOP iMPCs and FOP primary human muscle stem cells are used for modeling muscle stem cell engraftment and regeneration properties.

bi-potency compared to primary Hu-MuSCs, suggesting that iMPCs may not be as committed to the muscle lineage as primary adult muscle stem cells (*Xi et al., 2020*).

## Discussion

Developing optimal strategies for skeletal muscle regeneration and repair requires a detailed understanding of how these processes are regulated. Our study uses the abnormal ACVR1 activity of FOP to dissect how the BMP/ACVR1 pathways impact skeletal muscle repair (*Figure 7*). In the case of FOP, muscle injury can trigger disease progression or severe complications. Thus, generating large numbers of human iMPCs using transgene-free protocols holds promise for understanding pathological mechanisms and finding new therapeutic targets. The findings in FOP are also likely to help inform why HO sometimes forms in other conditions of muscle injury or inflammation, including burns, CNS injury, rheumatologic disorders, trauma, and surgery (*Matsuo et al., 2019*).

This study revealed several novel findings, including deficiencies in the muscle repair capacity of FOP Hu-MuSCs obtained from two different subjects, abnormal production of extracellular matrix components by iMSCs and FOP Hu-MuSCs that may contribute to FOP disease progression, and important differences between primary Hu-MuSCs and iMPCs that may impact other studies. In addition, our results suggest that there are muscle specific differences in FOP Hu-MuSC repair capacity, potentially explaining why some skeletal muscles appear to be protected from developing HO.

The iMPC and primary Hu-MuSC single cell transcriptomes revealed clear subsets of cells at all stages of muscle differentiation (quiescent, activated, and differentiated). Satellite cell subtype markers e.g. *COLs*, *DLK1*, *ID3*, and *HES1*, (*Barruet et al., 2020*) were also highly expressed in iMPCs suggesting that in vitro differentiation may favor specific subtypes of satellite cells. iMPCs also expressed high levels of mesenchymal/ECM markers, similar to that in human fetal muscle progenitor cells (*Xi et al., 2020*). These results suggest that iMPCs may retain a more progenitor-like phenotype as compared to Hu-MuSCs. In addition, we showed that PAX7+ iMPCs can engraft in muscle injury models in mice but at lower efficiency.

This iMPC system revealed that activation of the BMP pathway by the FOP *ACVR1R206H* mutation induced changes that could contribute to abnormal muscle healing. As expected, the classical SMAD pathways were active in FOP cells; however, it was unexpected to find that the p38 pathway via JUN/ FOS was more active in the FOP iMPCs, considering that FOP cultures had a higher proportion of cells

in the stem cell/progenitor and proliferating phases. While increased SMAD activity (*Billings et al., 2008*) or misinterpretation of the Activin A ligand (*Hatsell et al., 2015*) occur with the *ACVR1*[R206H] mutation, abnormal p38 signaling may be critical in some cell types like macrophages (*Barruet et al., 2018*). Since p38 is a major regulator of Hu-MuSC function (*Segalés et al., 2016*) and has been associated with inflammation and ECM accumulation in aged muscle (*Cosgrove et al., 2014*), further studies are needed to determine how this p38 signaling contributes to satellite cell subtype specification (*Barruet et al., 2020*) and affect healing in FOP.

Importantly, our iMPC and Hu-MuSC transplant studies showed no HO in the recipient mice, consistent with prior FOP genetic studies (*Dey et al., 2016*; *Lees-Shepard et al., 2018*) describing that other cell types may be directly responsible for bone formation. Indeed, several reports have suggested fibro/adipogenic progenitors (FAPs) as a candidate cell population to FOP pathogenesis. Interestingly, there are functional cross-talk between satellite cells and FAPs which is critical for muscle repair in homeostasis, disease, and aging contexts (*Biferali et al., 2019*; *Farup et al., 2015*; *Lukjanenko et al., 2019*; *Wosczyna and Rando, 2018*). However, FOP iMPCs showed increased chondrogenic/osteogenic and ECM gene expression, a feature seen in other bone-related cell types in FOP (*Barruet et al., 2016*; *Culbert et al., 2014*; *Lees-Shepard et al., 2018*). Thus, one possibility would be for FOP Hu-MuSCs to further contribute to HO formation indirectly by modulation of the osteogenic environment such as contributing to changes in muscle stiffness, as previously reported in FOP mice (*Haupt et al., 2019*; *Stanley et al., 2019*). This has also been reported in aged muscle, which show increased myofiber stiffness due to increased ECM deposition (*Hwang and Brack, 2018*). In addition, while our results suggest no major differences or possibly a slight increase in the ability to form skeletal muscle progenitors, activated ACVR1 by the *R206H* mutation decreased (but did not abrogate) in vitro formation of mature myoblasts of FOP hiPSCs and in vivo muscle repair after transplant of primary FOP Hu-MuSCs.

Comparing primary FOP Hu-MuSCs from muscles that develop HO (biceps) and non-affected muscle (diaphragm) suggests that the source of the Hu-MuSCs may impact engraftment efficiency. The re-injury model showed that engraftment of Hu-MuSCs from FOP biceps, but not from diaphragm, remained significantly decreased. Thus, it is intriguing to consider that the clinical sparing of the diaphragm from HO in patients with FOP may result from a less impaired or unimpaired muscle repair process in diaphragm satellite cells. Further delineation of muscle-specific Hu-MuSC properties will be revealing to understand this observation. Our finding that primary FOP Hu-MuSCs have lower engraftment ability provides a potential explanation for the poor skeletal muscle repair observed in patients with FOP (*Shore, 2012*). However, the FOP and control iMPCs showed no major differences in engraftment, possibly due to decreased assay sensitivity from the lower engraftment efficiency of iMPCs in general, which could be explained by iMPCs being more immature than primary Hu-MuSCs, the presence of cell contaminants after isolation, or that iMPCs represent a subtype of cells that may be more reflective of non-ossifying skeletal muscle like diaphragm.

This study has several limitations. Our transplant experiments used immunocompromised mice where mature B and T cells are absent and macrophages are defective (*Shultz et al., 2005*). This may dampen the engraftment/regenerative phenotypes, particularly as immune cells are important in muscle regeneration (*Furrer and Handschin, 2017*) and there is growing awareness of the role of the immune system in the pathogenesis of FOP (*Barruet et al., 2018*; *Convente et al., 2018*). Also, while our FACS strategy can isolate Hu-MuSCs and iMPCs without the need of transcription factors or markers, we noted that the engraftment efficiency still varied among the iMPC lines despite FACS purification. This heterogeneity is an ongoing problem among all published iMPC protocols to date. In our case, this was counter-balanced by our consistent findings across multiple lines and between iMPCs and the availability of primary Hu-MuSCs despite the absence of isogenic control and FOP lines. Although our primary cell studies were limited by the rarity of FOP (estimated at 1 in 1.4 million people) and even rarer suitable cadaveric samples where we were only able to obtain Hu-MSCs from two FOP subjects, the combination of iPSC-derived lineages with the rare primary samples provided multiple avenues for supporting our conclusions. Finally, the *ACVR1*[R206H] mutation increases BMP signaling but also introduces neofunction to Activin A. Our studies are not able to distinguish between these two contributing pathways. Our finding of increased p38 activity in the FOP iMPCs was also seen in FOP subject monocyte-derived macrophages (*Barruet et al., 2018*), suggesting that this alternate signaling pathway by ACVR1 should also be investigated further. Future studies, including the use of

isogenic control lines, testing of neutralizing antibodies to directly target the ACVR1$^{WT}$ or ACVR1$^{R206H}$ receptors, and the use of other small molecule inhibitors of pathways downstream of ACVR1 (e.g. p38 or SMAD), will help elucidate the different factors that contribute to the hiPSC-line specific effects, including potential roles for disease modifier genes or BMP pathway modulators.

This study shows that human iPSC-derived muscle progenitor cells can be a valuable tool to model musculoskeletal diseases of skeletal muscle injury and repair. Correlating the findings in iMPCs with primary Hu-MuSCs revealed an indirect role for skeletal muscle progenitors in HO formation, as well as potential subtypes of Hu-MuSCs that could contribute to skeletal muscle specific regenerative capacity (*Figure 7*). These studies highlight the importance of skeletal muscle regeneration in disease pathogenesis and establish a foundation for understanding how skeletal muscle repair and osteogenesis are linked.

# Materials and methods

**Key resources table**

| Reagent type (species) or resource | Designation | Source or reference | Identifiers | Additional information |
|---|---|---|---|---|
| Strain, strain background (*Mus musculus*) | NOD.Cg-Prkdcscid Il2rgtm1Wjl/SzJ | https://www.jax.org/strain/005557 | MGI:3577020 | 8–12 weeks old |
| Sequence-based reagent | Human RT-PCR Primers | Applied Biosystems Taqman Assays | *ACTB* Hs01060665_g1 | |
| Sequence-based reagent | Human RT-PCR Primers | Applied Biosystems Taqman Assays | *DYSTROPHIN* Hs007758098_m1 | |
| Sequence-based reagent | Human RT-PCR Primers | Applied Biosystems Taqman Assays | *PAX7* Hs00242962_m1 | |
| Sequence-based reagent | Human RT-PCR Primers | Applied Biosystems Taqman Assays | *CD56* Hs00941830_m1 | |
| Sequence-based reagent | Human RT-PCR Primers | Applied Biosystems Taqman Assays | *MYOD1* Hs00159528_m1 | |
| Sequence-based reagent | Human RT-PCR Primers | Applied Biosystems Taqman Assays | *MYF5* Hs00929416_g1 | |
| Sequence-based reagent | Human RT-PCR Primers | Applied Biosystems Taqman Assays | *MYOGENIN* Hs01072232_m1 | |
| Sequence-based reagent | Human RT-PCR Primers | Applied Biosystems Taqman Assays | *COL1A1* Hs01076780_g1 | |
| Sequence-based reagent | Human RT-PCR Primers | Applied Biosystems Taqman Assays | *ID1* Hs03676575_s1 | |
| Sequence-based reagent | Human RT-PCR Primers | Applied Biosystems Taqman Assays | *ID3* Hs00954037_g1 | |
| Antibody | Anti-Human DYSTROPHIN (mouse monoclonal) | DSHB | AB_2618157 | IF(1:10) |
| Antibody | Anti-Human/Mouse DYSTROPHIN (mouse monoclonal) | Thermofisher | AB_10978300 | IF(1:500) |
| Antibody | Anti-Human/Mouse PAX7 (mouse monoclonal) | DSHB | AB_528428 | IF(1:10) |
| Antibody | Anti-Human/Mouse MHC (mouse monoclonal) | DSHB | AB_2147781 | IF(1:100) |
| Antibody | Anti-LAMININ (rabbit polyclonal) | Sigma-Aldrich | AB_477163 | IF(1:250) |
| Antibody | Anti-Human SPECTRIN (mouse monoclonal) | Leica Microsystems | AB_442135 | IF(1:100) |

*Continued on next page*

*Continued*

| Reagent type (species) or resource | Designation | Source or reference | Identifiers | Additional information |
|---|---|---|---|---|
| Antibody | Anti-Human LAMIN A/C (mouse monoclonal) | Vector Laboratories | AB_2336546 | IF(1:100) |
| Antibody | Anti-Human CD31 (Beads) (Mouse monoclonal) | Miltenyi Biotec | 130-091-935 | 2 µl |
| Antibody | Anti-Human CD45 (Beads, mouse monoclonal) | Miltenyi Biotec | AB_2783001 | 5 µl |
| Antibody | Anti-Human CD31 AF450 (WM-59, mouse monoclonal) | Ebioscience | AB_10854276 | 5 µl |
| Antibody | Anti-Human CD34 eFluor450 (4H11, mouse monoclonal) | Ebioscience | AB_10733282 | 5 µl |
| Antibody | Anti-Human CD45 AF450 (30-F11, mouse monoclonal) | Ebioscience | AB_1518806 | 5 µl |
| Antibody | Anti-Human CD29 FITC (TS2/16, mouse monoclonal) | Ebioscience | AB_2043830 | 8 µl |
| Antibody | Recombinant human anti-CD56 APC-vio-770 (REA196) | Miltenyi Biotec | AB_2733136 | 8 µl |
| Antibody | Anti-Human CXCR4 PE (12G5, mouse monoclonal) | Ebioscience | AB_10669164 | 8 µl |
| Antibody | Anti-Human HNK1 PE (TB01, mouse monoclonal) | Ebioscience | AB_10804531 | 5 µl |
| Antibody | Anti-Human CD45 PE (30-F11, mouse monoclonal) | Ebioscience | AB_465668 | 5 µl |
| Antibody | Anti-Human CD31 PE (390, mouse monoclonal) | Ebioscience | AB_465632 | 5 µl |
| Antibody | Anti-Human CD56 APC (CMSSB, mouse monoclonal) | Ebioscience | AB_10854573 | 5 µl |
| Antibody | Anti-Human CXCR4 PE-Cy7 (12G5, mouse monoclonal) | Ebioscience | AB_1659706 | 5 µl |
| Antibody | FcR block | Miltenyi Biotec | AB_2892112 | 5 µl |
| Software, algorithm | GraphPad Prism | GraphPad Prism (https://graphpad.com) | SCr_002798 | |
| Software, algorithm | Seurat (3.1.5) | https://satijalab.org/seurat/ | SCR_007322 | SCR_021002 |
| Software, algorithm | Monocle (2.12.0) | http://cole-trapnell-lab.github.io/monocle-release/ | SCR_016339 | |
| Software, algorithm | cellranger | https://support.10xgenomics.com/single-cell-gene-expression/software/pipelines/latest/feature-bc | SCR_021002 | |
| Software, algorithm | FlowJo | https://www.flowjo.com | SCR_008520 | |

## Cell culture and differentiation

Pluripotent hiPSC lines derived from control and FOP fibroblasts (*Matsumoto et al., 2013*; *Spencer et al., 2014*) were cultured in mTeSR1 medium (StemCell Technologies) on irradiated SNL feeder cells (*McMahon and Bradley, 1990*) as described previously. hiPSCs were passaged at least once on Matrigel (Corning)-coated plates (150–300 µg/ml) to remove the SNLs before use in differentiation assays. ROCK inhibitor Y-27632 (10 µM, StemCell Technologies) was added to mTeSR1 when cells were split and removed the following day.

hiPSC lines were differentiated into skeletal muscle cells using modifications based on prior protocols (*Chal et al., 2016*; *Shelton et al., 2014*). Our hiPSC lines differentiated better with a lower cell number seeding and a longer time of recovery between the seeding and the start of the differentiation (2 days, data not shown). Cells were seeded at $7.5 \times 10^5$ cells per well of a 12-well plate on Matrigel two days before the differentiation medium (E6 medium, supplemented with either 10 µM CHIR99021 (Tocris) for 2 days or with 3 µM CHIR99021 and 0.5 µM LDN193189 for six days). Cells were then

grown in un-supplemented E6 media until day 12, then changed to StemPro-34 media supplemented with 10 ng/ml bFGF until day 20. The medium was then replaced by E6 medium until day 35, when DMEM/F12 supplemented with N2 (Gibco) and Insulin-Transferrin-Selenium (ITS-A, 100 X Gibco) was added. Media was changed daily until harvest at day 50. After sorting, cells were plated in satellite cell media [DMEM/F12 (Gibco), 20 % FBS (Hyclone), 1 X ITS (Gibco), 1 X Penicillin/ Streptomycin (Gibco)] for further functional assays (*Figure 2A and F*). Once cells reached confluence, cells were cultured in differentiation media (DMEM, 2 % horse serum, 1 X Penicillin/ Streptomycin; Gibco).

## Cell lines

Pluripotent hiPSC lines were previously derived from the following control and FOP fibroblasts (*Matsumoto et al., 2013*; *Spencer et al., 2014*): FF-WT-BJ (Foreskin, Stemgent 08–0027), HDF-WT-1323 (Fibroblast, Cell Applications 1323), HDF-WTc (Skin biopsies obtained from donors), HDF-FOP1 (Skin Fibroblast, Corrielle GM00513), HDF-FOP2 (Skin Fibroblast, Corrielle GM00783), and HDF-FOP3 (Skin biopsies obtained from donors). All hiPSCs are tested periodically for mycoplasma and have been negative to date. Cell lines are yearly authenticated using DNA fingerprinting.

## Flow cytometry

HNK1⁻CD45⁻CD31⁻CXCR4⁺CD29⁺CD56^dim cells were sorted from skeletal muscle differentiation of control and FOP hiPSCs. Human primary satellite cells were isolated and sorted as described (*Garcia Steven et al., 2018*; *Garcia et al., 2017*). Human muscle was freshly harvested and stored in DMEM with 30 % FBS at 4 °C overnight or for two extra days (delay due to shipping). Muscle samples were digested, erythrocytes were lysed, and hematopoietic and endothelial cells were depleted with magnetic column depletion using CD31, CD34, and CD45 (eBioscience). Cells were further gated as described in *Figure 1—figure supplement 1A* and sorted for CXCR4⁺/CD29⁺/CD56⁺ and collected for subsequent experimentation.

For the staining, cells were treated with Accutase for 20 min at 37 C, washed with FACS buffer, and stained for HNK1-PE, CD45-PE, CD31-PE, CD56-APC, CXCR4-PE-Cy7, and CD29-FITC (eBioscience). HNK1⁻CD45⁻CD31⁻ cells [to select against neuronal cells (HNK1, Human Natural Killer-1) (*Choi et al., 2016*; *Hicks et al., 2018*), hematopoietic cells (CD45), and endothelial cells (CD31)], co-expressing CD29, CXCR4, and intermediate CD56, markers present on human PAX7⁺ cells (*Garcia Steven et al., 2018*; *Garcia et al., 2017*; *Xu et al., 2015*) were sorted with a FACSArialII (BD Biosciences) and Sytox Blue (Life Technologies) was used as a viability marker. Alternatively, cells were permeabilized and fixed (Fix/Perm Buffer Set, BioLegend). Fixed cells were first incubated with primary antibodies PAX7 (DSHB), CD29 (BD Biosciences), and CD56 (BD Biosciences) following by secondary antibodies (Alexa350-conjugated goat anti-mouse IgG, Alexa488-conjugated goat anti-rat IgG and Alexa546-conjugated donkey anti-goat IgG, Life Technologies).

## Animal care and transplantation studies

All mouse studies were performed using protocols approved by the UCSF Institutional Animal Care and Use Committee. Mice were either bred and housed in a pathogen-free facility at UCSF or purchased from The Jackson Laboratory. Eight to 12-week-old NSG (NOD.Cg-Prkdcscid Il2rgtm1Wjl/ SzJ) mice were randomized to all experimental groups by sex and littermates. Each mouse was irradiated with 18 Gy before transplantation. Isolated primary human satellite cells (Hu-MuSCs) or iMPCs were injected with 50 µl 0.5 % bupivacaine directly into the tibialis anterior (TA) muscle of one leg as described (*Garcia Steven et al., 2018*; *Garcia et al., 2017*). The TA for each mouse was harvested at week 5 or week 10 after transplantation and frozen in O.C.T. compound in 2-methylbutane chilled in liquid nitrogen. Serial 6 µm transverse frozen sections were analyzed or stored at –80 °C.

Collected tibialis anterior frozen cross sections were fixed in 4 % PFA for 10 min at room temperature, washed with PBST PBS with 0.1 % Tween-20 (Sigma-Aldrich), and blocked in PBS with 10 % goat serum for 1 hr at room temperature. Slides were then incubated 4 hrs at room temperature with the following: mouse monoclonal anti-human DYSTROPHIN (DSHB), mouse monoclonal IgG1 anti-PAX7 (DSHB), rabbit polyclonal anti-Laminin (Sigma-Aldrich), mouse monoclonal IgG2b anti-human SPECTRIN (Leica Microsystems), and mouse monoclonal IgG2b anti-human LAMIN A/C (Vector Laboratories). After PBST wash, slides were incubated with the following secondary antibodies: Alexa Fluor 555 goat anti-mouse IgG, Alexa Fluor 594 goat anti-mouse IgG1, Alexa Fluor 488 goat anti-mouse

IgG2b, and Alexa Fluor 647 goat anti-rabbit (Life Technologies). Finally, sections were mounted with VECTASHIELD mounting media with DAPI (Vector Laboratories). All samples were examined using a Leica upright or DMi8 Leica microscope. Sections with the most human fibers were used for human DYSTROPHIN and PAX7 quantification for each condition, as described below.

Cell Immunostaining and NSG Tibialis Anterior Analysis iMPCs were fixed with 4%PFA/PBS for 10 min at room temperature, permeabilized with 0.1%Triton-100X (Sigma-Aldrich), and blocked with 5 % BSA (Sigma-Aldrich). Cells were stained overnight with primary antibodies for PAX7, MYOGENIN, DYSTROPHIN, and MHC. Cells were then incubated for 1 hr at room temperature in the dark with secondary antibodies Alexa488-conjugated goat anti-mouse IgG and Alexa546-conjugated goat anti-mouse IgG (Invitrogen). Nuclei were stained with DAPI (Sigma-Aldrich). Images were taken on a Nikon Eclipse E800 or Leica DMI 4000B.

## Immunohistochemistry and immunofluorescence of human muscle samples

Human muscle samples were fixed in neutral buffered formalin for 24 hr and then placed in 70 % ethanol for at least 24 hr. The sample with heterotopic bone was decalcified in 10 % EDTA (pH 7.2–7.4) before paraffin embedding and sectioning. Sections were stained with hematoxylin and eosin (J. David Gladstone Institutes Histology Core) or for alcian blue (pH 1.0) for cartilage and nuclear red stain for nuclei.

Freshly harvested human muscle was stored in DMEM with 30 % FBS at 4 °C, or snap in frozen in O.C.T. compound in 2-methylbutane chilled in liquid nitrogen. Serial 6 μm transverse frozen sections were analyzed or stored at –80 °C and processed similarly to the mouse TA samples above. Sections were stained with PAX7 (DSHB) and mouse monoclonal anti-Collagen Type I (Millipore-Sigma). Details about specimens are in the *Figure 1—source data 1*.

## RT-PCR and quantitative analysis

Tissues were collected in TRI Reagent (Sigma-Aldrich) to isolate total RNA using the Arcturus PicoPure RNA kit (Applied Biosystems) as previously described for small samples (*Schepers et al., 2012*). A total of 0.2–0.5 μg of RNA were transcribed into cDNA with VeriScript cDNA synthesis kit (Affymetrix). cDNA was then pre-amplified with GE PreAmp Master Mix (Fluidigm Inc). Real-time quantitative PCR was performed in triplicated with either VeriQuest Probe qPCR Master Mix (Affymetrix) or Taqman Universal PCR Master Mix (Life Technologies) on either a Viia7 thermocycler (Life Technologies) or on a BioMark 48.48 dynamic array nanofluidic chip (Fluidigm, Inc) according to manufacturers' instructions. *ACTB* was used for normalization as endogenous control.

Single cell RNA Sequencing and Analysis scRNAseq was performed using the Chromium Single Cell 3' Reagent Version 2 Kit from 10 X Genomics. 45,000 (FOP) and 30,000 (control) HNK1⁻CD45⁻CD31⁻ CXCR4⁺CD29⁺CD56$^{dim}$ cells isolated from the iMPC differentiations were loaded onto one well of a 10X chip to produce Gel Bead-in-Emulsions (GEMs). GEMs underwent reverse transcription to barcode RNA before cleanup and cDNA amplification. Libraries were prepared with the Chromium Single Cell 3' Reagent Version 2 Kit. Each sample was sequenced on 1 lane of the NovaSeq 6,000 S4. Sequencing reads were processed with Cell Ranger version 2.0.0. using the human reference transcriptome GRCh38. The estimated number of cells, mean reads per cell, median genes per cells, median UMI (Unique Molecular Identifier) counts per cells as well as other quality control information are summarized in *Figure 4—source data 1*. Gene-barcoded matrices were analyzed with the R package Seurat v3.1.5 (*Satija et al., 2015*; *Stuart et al., 2019*; *Team, 2014*; *Zheng et al., 2017*). Gene core matrices from single cell RNA sequencing of primary human satellite cells isolated from a vastus muscle (*Barruet et al., 2020*) was used when comparing the transcriptional profile of hiPS-derived HNK1⁻CD45⁻CD31⁻ CXCR4⁺CD29⁺CD56$^{dim}$ cells. For the comparison with primary Hu-MuSCs, hiPSC-derived cell sequencing reads were re-aligned using the human reference transcriptome hg19. Cells with fewer than 500 genes, greater than 5,000 genes and genes expressed in fewer than five cells were not included in the downstream analyses. Cells with more than 10 % mitochondrial counts were filtered out. Samples were normalized with NormalizeData using default settings. The FindVariableFeatures function was used to determine subset of feature that exhibit high cell-to-cell variation in each dataset based on a variance stabilizing transformation ('vst'). We used the default setting returning 2000 feature per dataset. These were used for downstream analysis. In the case of the

merged data analysis samples were combined utilizing the FindIntegrationAnchors function with the 'dimensionality' set at 30. Then, we ran these 'anchors' to the IntegratData function for batch correction for all cells enabling them to be jointly analyzed. The resulting outputs were scaled mitochondrial contamination regressed out with the ScaleData function. In addition, while we didn't regress out heterogeneity associated with cell cycle stage since it is an important factor in determining the state of quiescence of our sorted human muscle stem cells, we regressed out differences between G2/M and S cell cycle stage. PCA was performed with RunPCA, and significant PCs determined based on the Scree plot utilizing the function PCElbowPlot. The resolution parameter in FindClusters was adjusted to 0.5. Clusters were visualized by UMAP with Seurat's RunUMAP function. We performed differential gene-expression utilizing Seurat v3's FindMarkers function with default settings which utilizes the Wilcoxon rank-sum test to calculate adjusted p values for multiple comparisons. We used the Cell-CycleScoring function to assign score based on the expression of G2/M and S phase markers (*Regev et al., 2017*). Myogenic cells were further analyzed by sub-clustering using the subset function. We then use the FindNeighbors (dims = 15) and FindClusters (resolution = 0.4) functions on the myogenic cell subset of the merged hiPSCs samples only to identify sub-clusters corresponding to different myogenic states. To order the cells in pseudotime based on their transcriptional similarity we used Monocle 2.12. Variable genes from Seurat analysis were used as input and clusters were projected onto the minimum spanning tree after ordering. Gene expression patterns were plotted with plot_genes_branched_heatmap, plot_genes_branched_pseudotime, and plot_multiple_branches_pseudotime. The BEAM (branch expression analysis modeling) function was used to score gene significance in a branch-dependent manner. Cells were re-ordered using the orderCells function to set branch A (myocytes) in *Figure 6H* as the 'root-state'. This allowed us to determine genes that were significantly branch dependent in branch A (mainly Hu-MuSCs) vs branch B (hiPS-derived cells) using the BEAM analysis.

## Statistical analysis

The data were analyzed utilizing GraphPad Prism v.7 software (GraphPad) using one-way (transplant) and two-way ANOVA with post hoc Tukey's or Sidak's multiple comparison test (gene expression). The Sidak test was used when comparing means between control and FOP, and the Tukey test was used when means of both control and FOP were compared together with other groups for the gene expression data. For the transplantation studies, at least three mice were used per group. At least three biological replicates were performed for each experiment unless indicated otherwise. All error bars are depicted as standard deviation, p-values are (*p < 0.05, **p < 0.01, ***p < 0.001, ****p < 0.0001).

## Human specimen procurement

Human samples were collected through the UCSF Biospecimens and Skeletal Tissues for Rare and Orphan Disease Genetics (BSTROnG) Biobank, using protocols approved by the UCSF Institutional Review Board. All participants provided written consent. For the transplantation studies (*Figure 1D–H* and *Figure 1—figure supplement 1*), biopsy of the control subject was obtained from a 44 yo female healthy individual undergoing surgery at UCSF, and the muscle from the FOP patient was obtained at autopsy from a 55 yo female. Written informed consent was obtain from all subjects or their families.

## Acknowledgements

The authors thank Kelly Wentworth and Samuel Kou for their assistance collecting the autopsy samples and Francesco Tedesco for his technical support and discussion on the AFM grant. This work was supported by a NIH/NIAMS R01AR066735 to ECH, a French Muscular Association (AFM-Telethon) Trampoline grant to ECH and EB, the Radiant Hope Foundation to ECH, and the UCSF Cohort Development Grant to ECH; the California Institute for Regenerative Medicine Fellowship Program to UCSF (TG2-01153) to EB, and the UCSF Program for Breakthrough Biomedical Research (PBBR) to EB; and the NIH R01AR072638-03 to JHP. Finally, the authors would like to thank the patients and their families for their generous specimen donations.

## Additional information

### Competing interests

Edward C Hsiao: ECH receives clinical trial research funding from Clementia Pharmaceuti-cals, an Ipsen company, and Neurocrine Biosciences, Inc., through his institution. ECH received prior funding from Regeneron Pharmaceuticals, through his institution. ECH serves in an unpaid capacity on the international FOP Association Medical Registry Advisory Board, on the International Clinical Council on FOP, and on the Fibrous Dysplasia Foundation Medical Advisory Board. These activities pose no conflicts for the presented research. The other authors declare that no competing interests exist.

### Funding

| Funder | Grant reference number | Author |
|---|---|---|
| National Institute of Arthritis and Musculoskeletal and Skin Diseases | R01AR066735 | Edward C Hsiao |
| French Muscular Dystrophy Association | Trampoline grant | Emilie Barruet Edward C Hsiao |
| Radiant Hope Foundation | | Edward C Hsiao |
| University of California, San Francisco | UCSF Cohort Development Grant | Edward C Hsiao |
| California Institute for Regenerative Medicine | TG2-01153 | Emilie Barruet |
| University of California, San Francisco | UCSF Program for Breakthrough Biomedical Research | Emilie Barruet |
| National Institute of Arthritis and Musculoskeletal and Skin Diseases | R01AR072638-03 | Jason H Pomerantz |

The funders had no role in study design, data collection and interpretation, or the decision to submit the work for publication.

### Author contributions

Emilie Barruet, Conceptualization, Data curation, Formal analysis, Funding acquisition, Investigation, Methodology, Project administration, Software, Validation, Visualization, Writing – original draft, Writing – review and editing; Steven M Garcia, Investigation, Validation; Jake Wu, Blanca M Morales, Stanley Tamaki, Investigation; Tania Moody, Data curation, Software; Jason H Pomerantz, Clinical samples, Clinical samples, Conceptualization, Funding acquisition, Resources, Supervision, Writing – review and editing; Edward C Hsiao, Clinical samples, Clinical samples, Conceptualization, Funding acquisition, Project administration, Resources, Supervision, Writing – original draft, Writing – review and editing

### Author ORCIDs

Emilie Barruet http://orcid.org/0000-0002-4593-024X
Steven M Garcia http://orcid.org/0000-0002-7833-6677
Jason H Pomerantz http://orcid.org/0000-0002-5107-1883
Edward C Hsiao http://orcid.org/0000-0001-8924-106X

### Ethics

Human subjects: Human samples were collected through the University of California - San Francisco Biospecimens and Skeletal Tissues for Rare and Orphan Disease Genetics (BSTROnG) Biobank, using protocols approved by the UCSF Institutional Review Board (10-03053 and 11-06711). All participants provided written consent, which includes consent to collect, use, and publish research data.

All mouse studies were performed using protocols approved by the UCSF Institutional Animal Care and Use Committee. All of the animals were handled according to approved institutional animal care and use committee (IACUC) protocols (AN181101-02A) of the University of California, San Francisco.

### Decision letter and Author response

Decision letter https://doi.org/10.7554/eLife.66107.sa1
Author response https://doi.org/10.7554/eLife.66107.sa2

---

## Additional files

### Supplementary files

• Transparent reporting form
• Source code 1. hiPSCs samples integration.
• Source code 2. Myogenic sub-clustering.
• Source code 3. Pseudotime analyses.
• Source code 4. Primary Hu-MuSCs/hiPSCs integration.

### Data availability

Sequencing data have been deposited in GEO under accession codes GSE151918. All data generated or analysed during this study are included in the manuscript and supporting files. Source data files have been provided for Figures 4,5,6. The dataset used for the primary Hu-MuSCs can be found here: https://doi.org/10.7272/Q65X273X. Detailed scripts can be found at https://github.com/EmilieB12/FOP_muscle/tree/main (copy archived at https://archive.softwareheritage.org/swh:1:rev:2c30ff4ef2608c0a44360047e54b9ad2d3017d0c) or in Source code 1–4.

The following dataset was generated.

| Author(s) | Year | Dataset title | Dataset URL | Database and Identifier |
|---|---|---|---|---|
| Barruet E, Garcia SM, Jake Wu J, Morales BM, Stanley Tamaki S, Moody T, Pomerantz JH, Hsiao EC | 2020 | Control and FOP Human iPSC derived myogenic cell single cell RNA sequencing | https://www.ncbi.nlm.nih.gov/geo/query/acc.cgi?acc=GSE151918 | NCBI Gene Expression Omnibus, GSE151918 |

The following previously published datasets were used:

| Author(s) | Year | Dataset title | Dataset URL | Database and Identifier |
|---|---|---|---|---|
| Pomerantz JH, Barruet E | 2020 | Functionally heterogeneous human satellite cells identified by single cell RNA sequencing | https://doi.org/10.7272/Q65X273X | Dryad Digital Repository, 10.7272/Q65X273X |

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
