## [Editor Report]

The manuscript by Barruet et al., investigates an interesting and rare skeletal muscle dystrophy (FOP). They use both primary and induced (iPSC) muscle stem cells to determine how regeneration/engraftment is affected in this condition. The authors use a blend of histology and transcriptional approaches to determine ECM remodeling and myogenic capacity from both cell/tissue lines. The experiments are well conducted and use strong approaches and statistical measures to test their hypothesis. Overall, this is a quality manuscript on a rare muscle disease that will establish a novel model to study FOP and initial data to elucidate the molecular pathology of HO.

---

## [Decision Letter]

**Decision letter after peer review:**

Thank you for submitting your article "*ACVR1*^*R206H*^ increases osteogenic/ECM gene expression and impairs myofiber formation in human skeletal muscle stem cells" for consideration by *eLife*. Your article has been reviewed by 3 peer reviewers, and the evaluation has been overseen by a Reviewing Editor and Kathryn Cheah as the Senior Editor. The reviewers have opted to remain anonymous.

Essential revisions:

1. A major concern for this study is that there is a small sample size herein and it was felt that the statistical analysis was lacking. It is understood that FOP is a rare genetic disease and obtaining a patient sample is extremely difficult. However, all of the data in Figure 1 is obtained from two patients. Conclusions drawn from an n=2 need to be appropriately circumspect.

2. In the transplantation experiments, several samples had zero engraftments while other samples had extremely low engraftment even in the control group less than 0.1% of cells were engrafted. If you exclude non-engrafted recipients, all of the results are inconclusive. This issue was raised by all of the reviewers.

3. Some of the representative images do not really show what the authors are claiming. I am not sure whether this is due to low-resolution images or not but it is difficult to see whether these are really expressing myogenic proteins or if they are false positives. Moreover, in Figure 2E, there are more samples that do not express PAX7 and MYOGENIN than cells that do. Without reaching statistical significance, authors cannot claim their conclusion that hiPSC have differentiated into MuSC like cells. Furthermore, authors should have tested other myogenic genes such as MYOD, MYF5, MRF. DYSTROPHIN was not a choice of protein for immunofluorescence.

4. Compared to fibrosis and lipid infiltration, heterotopic ossification in muscle is a rare incidence that only occurs following traumatic injury or limb amputations. Even FOP is a rare genetic disorder. Thus, authors should not generalize that HO is a common abnormality that follows a lack of muscle repair or regeneration.

5. There are several reports that suggest fibrogenic/adipogenic progenitors aberrantly differentiate into HO phenotype. The authors fail to discuss the importance of FAPs in adult myogenesis and muscle stem cell niche as potential confounding factors altering BMP signaling in FOP or HO.

6. The introduction must be re-written to clarify the rationale and objectives of the study. It should be re-focused around the mechanisms of FOP and synthesize previous data on FOP and the cell types involved. There is confusion between heterotopic ossification and FOP as these conditions may cause ectopic bone formation in skeletal muscle through independent mechanisms. The rationale for manipulating BMP signaling for generating hiMPCs is also unclear and maybe not justified.

7. The title is misleading and should be edited to include establishing a model, not highlighting the functional consequences of the gene mutation, which I believe are clinically implied.

8. Figure 2 should include quantitative data and higher quality images. One limitation is the lack of isogenic control for the FOP lines. This limitation should be mentioned in the discussion.

9. Figure 3 should include FMO control or non-labelled cells. In panel B, statistical analyses should be performed on data from biological replicates only and not biological + technical replicates.

10. Figure 6: What are the identities of non-myogenic clusters?

11. If the authors want to imply function, are there inhibitors to this pathway that could serve to reverse the pathology in this model? I see BMP inhibitors are used however, targeting upstream (i.e. effecting the activin receptor itself) mechanisms would serve as a more direct proof of concept experiment.

12. It would be interesting if the BMP or activin inhibitor would affect transplantation efficiency. From what I can tell, the LDN inhibitor was only used in vitro.

13. Results show low efficiency of myogenic differentiation from hiMPCs and variable engraftment capacity that is not affected by ACVR1R206H mutation.

14. The authors conclude that the abnormal expression of ECM and chondro/osteogenic markers by FOP iMPC may modulate the niche environment and may favor ectopic ossification. This conclusion thus requires stronger results to support it.

For the scRNAseq analyses presented in Figures 4 and 5:

15. Overall, the reviewers found this section interesting, but there were many flaws/concerns raised with the analysis conducted. Some interpretations of scRNAseq are not fully supported by the results. The cells of interest (skeletal muscle stem cells) appear very diluted in the dataset. The expression of ECM and chondro/osteogenic markers by FOP iMPCs is not convincing and more specific chondro/osteogenic markers should be analyzed.

16. Fewer cells were analysed in the control sample compared to FOP sample. The authors should indicate whether analyses were performed to compensate for differences in the sequencing depth.

17. In Figure 4, very few Pax7+ and Myod + cells are detected in the myogenic cluster. To what extent are the differences between control and FOP samples due to non pax7+ cells?

18. The non-myogenic clusters should be better characterized in Figure 4. The myogenic clusters should also be better defined in Figure 5, especially clusters 0 and 2.

19. In Figure 5, cluster # should be specified in panel B.

20. The pseudotime analyses are were very confusing to the reviewers and the suitability and rigor of this analysis was questioned. The starting point A contains cluster 4 cells that are MYOG+ and MYOD1+. In panel D, how can differentiation markers exhibit a non-linear progression? Chondro/osteogenic markers should be validated and more specific.

*Reviewer #1:*

The study by Barruet et al., reports interesting observations on muscle stem/satellite cells derived from fibrodysplasia ossificans progressiva (FOP) patients, both hiPSC differentiated into SC-like cells and primary cells. The authors performed a variety of transplantation/histological analyses, as well as transcriptional profiling. Weaknesses were noted with the data analysis and in places, claims on mechanism were overly strongly stated.

*Reviewer #2:*

Barruet et al., explore the impact of ACVR1R206H mutation responsible for FOP on skeletal muscle stem cells. In this rare genetic disease, over-activating point mutation in ACVR1 causes ectopic bone formation within skeletal muscles. This pathological response may occur due in part to abnormal skeletal muscle repair following injury, yet the cellular bases are not fully understood. Previous studies have revealed that the cells of origin for the ectopic bone formation are mesenchymal populations within the skeletal muscle interstitium, the fibro-adipo-progenitors (FAP), while ACVR1R206H mutation in other cell types such as endothelial cells or muscle stem cells alone do not cause the disease. However, the ACVR1R206H mutation in cell types other than FAP may contribute to the abnormal tissue environment and to the progression of the disease. Research aiming to elucidate the cellular mechanisms of FOP is hampered by the difficulty to have access to patient samples as muscle biopsy itself can trigger an injury response contributing to disease progression. Dr. Hsiao's team has developed over the past years the iPSCs technology to obtain FOP iPSCs in order to assess the consequences of ACVR1R206H mutation in several cell lineages. In this study, the authors concentrate on the myogenic lineage, and compare the muscle regenerative potential and transcriptome profiling by scRNAseq of iPSC-derived muscle stem/progenitor cells (iMPCs) and primary human muscle stem cells (Hu-MuSCs). Overall the data is of high quality. The results reveal that Hu-MuSCs from FOP patients have reduced regenerative potential compared to control. Reduced engraftment capacity was observed for Hu-MuSCs isolated from biceps but not diaphragm muscle. This correlates with the fact that diaphragm muscle is less affected in FOP, although results from diaphragm control would have further supported the claim. The authors used iPSCs from FOP patients and control to perform scRNAseq analyses. Reporting the similarities and differences in transcriptional signature of primary Hu-MuSCs and hiMPCs is an important contribution to the field and validates the use of iMPCs to further elucidate disease mechanisms. The conclusions drawn from the data presented were overreaching, which was in part due to a lack of clarity in the stated problem statement they were trying to address.

*Reviewer #3:*

The manuscript by Barruet et al., investigates an interesting and rare skeletal muscle dystrophy (FOP). They use both primary and induced (iPSC) muscle stem cells to determine how regeneration/engraftment is affected in this condition. The authors use a blend of histology and transcriptional approaches to determine EMC remodeling and myogenic capacity from both cell/tissue lines. The experiments are well conducted and use strong approaches and statistical measures to test their hypothesis. Overall, this is a quality manuscript on a rare muscle disease that will establish a novel model to study FOP and initial data to elucidate the molecular pathology of HO.

---

## [Author Response]

Essential revisions:1. A major concern for this study is that there is a small sample size herein and it was felt that the statistical analysis was lacking. It is understood that FOP is a rare genetic disease and obtaining a patient sample is extremely difficult. However, all of the data in Figure 1 is obtained from two patients. Conclusions drawn from an n=2 need to be appropriately circumspect.

We agree that the small sample size is a major limitation of our study, specifically in Figure 1. However, the ability to collect more samples is hampered by the rarity of FOP, and even more infrequent donations of autopsy material suitable for our studies particularly in the setting of COVID19. We clarify this limitation throughout our manuscript (including within the limitations section of our discussion), and note this limitation in the conclusion for Figure 1 accordingly.

2. In the transplantation experiments, several samples had zero engraftments while other samples had extremely low engraftment even in the control group less than 0.1% of cells were engrafted. If you exclude non-engrafted recipients, all of the results are inconclusive. This issue was raised by all of the reviewers.

In our study, the engraftment efficiency of the sorted human primary control satellite cell is comparable to previously published reports (Xu et al., 2015 Stem Cell Reports; Garcia et al., 2018 Stem Cell Reports). We agree that the low engraftment of our iMPCs poses challenges for our analysis. We explore several reasons for this, including the possibility of other cell types present in the cultures (as described in Figure 4B-C) such as neuroprogenitor, neuroepithelial, glial, and mesenchymal cells. In addition, we believe that inclusion of the low engraftment efficiency samples in our data analyses is important, since we did not know if the ACVR1 mutation might increase engraftment. Such an unexpected effect was previously observed, where the *ACVR1 ^R206H^* mutation unexpectedly increased efficiency of iPSC formation (Hayashi, et al., PNAS, 2016). We revised the discussion to better state this limitation, and adjusted text throughout the manuscript to clarify this limitation.

3. Some of the representative images do not really show what the authors are claiming. I am not sure whether this is due to low-resolution images or not but it is difficult to see whether these are really expressing myogenic proteins or if they are false positives. Moreover, in Figure 2E, there are more samples that do not express PAX7 and MYOGENIN than cells that do. Without reaching statistical significance, authors cannot claim their conclusion that hiPSC have differentiated into MuSC like cells. Furthermore, authors should have tested other myogenic genes such as MYOD, MYF5, MRF. DYSTROPHIN was not a choice of protein for immunofluorescence.

Thank you for identifying these concerns. We included new images with higher resolution that better show the expression of myogenic proteins in Figure 2. We now include images showing MYOGENIN staining in addition to PAX7 and DYSTROPHIN (Figure 2B,C). In Figure 2D, the heterogeneity of our myogenic gene expression data can be explained by the presence of other cells types which has been described previously (Shelton et al., 2015 Stem Cell Reports; Hicks et al., 2018 Nat Cell Biol) and shown in Figure 2—figure supplement 1B with TUJ-1 staining, Figure 3, and Figure 3—figure supplement 1A with FACS analysis for CD45 (hematopoietic cells) and CD31 (endothelial cells). We believe our new images better show our conclusion that hiPSCs have differentiated into hMuSC-like cells. We also added the gene expression analysis by qPCR of *MYOD1* and *MYF5* to support our findings.

4. Compared to fibrosis and lipid infiltration, heterotopic ossification in muscle is a rare incidence that only occurs following traumatic injury or limb amputations. Even FOP is a rare genetic disorder. Thus, authors should not generalize that HO is a common abnormality that follows a lack of muscle repair or regeneration.

We agree that fibrosis and lipid infiltration are important signs of muscle damage. However, heterotopic ossification is a surprisingly common event. In addition to traumatic injury or limb amputations, heterotopic ossification also affects patients with burns, CNS injury, rheumatologic disorders, and surgery (Matsuo et al., 2019 Curr Osteoporos Rep; Agarwal et al., 2016 PNAS; Meyers et al., 2019 JBMR Plus; Sullivan et al., 2013 Bone Joint Res), and more recently as a rare but significant COVID-19 complication (Aziz et al., 2021 Radiol Case Rep).

We revised the introduction and discussion to include this information for better context, and also made minor changes throughout the manuscript to indicate that HO is an uncommon but significant complication of muscle injury.

5. There are several reports that suggest fibrogenic/adipogenic progenitors aberrantly differentiate into HO phenotype. The authors fail to discuss the importance of FAPs in adult myogenesis and muscle stem cell niche as potential confounding factors altering BMP signaling in FOP or HO.

We expanded our discussion of FAPs in the introduction and discussion, and how their cross-talk with satellite cells may impact the pathogenesis of FOP or HO in general.

6. The introduction must be re-written to clarify the rationale and objectives of the study. It should be re-focused around the mechanisms of FOP and synthesize previous data on FOP and the cell types involved. There is confusion between heterotopic ossification and FOP as these conditions may cause ectopic bone formation in skeletal muscle through independent mechanisms. The rationale for manipulating BMP signaling for generating hiMPCs is also unclear and maybe not justified.

The introduction has been reorganized to help clarify the rationale and objectives. We also clarified our rational for examining how manipulating BMP signaling impacts generating Hu-MuSC like cells.

7. The title is misleading and should be edited to include establishing a model, not highlighting the functional consequences of the gene mutation, which I believe are clinically implied.

We have accepted the modified title. Thank you for this suggestion. “Modeling the *ACVR1*^*R206H*^ mutation in human skeletal muscle stem cells”

8. Figure 2 should include quantitative data and higher quality images. One limitation is the lack of isogenic control for the FOP lines. This limitation should be mentioned in the discussion.

Higher quality images have been added to Figure 2. Since immunohistochemistry is a semi-quantitative assay and significantly impacted by the staining efficiency and specificity of the antibodies, we believe the relative expression assessments found in Figure 2D, which are based on real time qPCR, provide a more accurate quantitative assessment.

The lack of isogenic lines and its consequent limitation are included in the discussion and the limitations section.

9. Figure 3 should include FMO control or non-labelled cells. In panel B, statistical analyses should be performed on data from biological replicates only and not biological + technical replicates.

Non-labeled cells are now included in the Figure 3—figure supplement 1C. Statistical analyses are now performed on biological replicates and are represented in Panel B of Figure 3.

In panel B, statistical analyses have been performed on 6 biological replicates using a two-way ANOVA Tukey test where the mean of each cell line was used. All 6 cell lines, with their own technical replicates, are shown in Figure 3B. We also performed statistical analyses on Ctrl vs FOP for each sort groups and found no significance.

10. Figure 6: What are the identities of non-myogenic clusters?

The identities of non-myogenic clusters are now clearly labelled in Figures 6B and C.

11. If the authors want to imply function, are there inhibitors to this pathway that could serve to reverse the pathology in this model? I see BMP inhibitors are used however, targeting upstream (i.e. effecting the activin receptor itself) mechanisms would serve as a more direct proof of concept experiment.

Thank you for suggesting those experiments in which we could better tease out the mechanism by using upstream inhibitors such as activin A neutralizing antibodies. As discussed in our manuscript, the number of primary muscle stem cells we can obtain from FOP patients is extremely limited, and so we are unable to test such inhibitors in vivo using these primary cells. In addition, strategies to directly target *ACVR1*^*WT*^ vs. *ACVR1*^*R206H*^ receptors are still being developed. Also, the direct targeting of the BMP pathway b the inhibitors will also inhibit the aberrant signaling induced by Activin A on ACVR1^R206H^. We hope to examine these intriguing questions in the future, using our human iPSC models and novel pathway inhibitors. This is clarified in the limitations paragraph of the discussion.

12. It would be interesting if the BMP or activin inhibitor would affect transplantation efficiency. From what I can tell, the LDN inhibitor was only used in vitro.

We agree that this would be a very interesting question to pursue. LDN has a long track record of being used to inhibit the SMAD signaling pathways of the BMPs, and particularly the signaling downstream of ACVR1, and in our results suggested benefits for both FOP and control cells in myogenic differentiation. However, the heterogeneity in our current studies, the relatively low efficiency of transplantation by both control and FOP cells, and the lack of large numbers of primary cells prevent us from dissecting further the other signaling circuitry of ACVR1 (ie SMAD vs. p38, or ACVR1 itself). We hope to pursue these detailed studies in the future. This is clarified in the limitations paragraph of the discussion.

13. Results show low efficiency of myogenic differentiation from hiMPCs and variable engraftment capacity that is not affected by ACVR1R206H mutation.

This is a current limitation of our system, and affects both control and *ACVR1*^*R206H*^ cells. We include a more detailed discussion of the low engraftment efficiency in our discussion.

14. The authors conclude that the abnormal expression of ECM and chondro/osteogenic markers by FOP iMPC may modulate the niche environment and may favor ectopic ossification. This conclusion thus requires stronger results to support it.

We have revised our Results section 7 to include additional explanation of our rationale, and also added additional information in the discussion.

For the scRNAseq analyses presented in Figures 4 and 5:

15. Overall, the reviewers found this section interesting, but there were many flaws/concerns raised with the analysis conducted. Some interpretations of scRNAseq are not fully supported by the results. The cells of interest (skeletal muscle stem cells) appear very diluted in the dataset. The expression of ECM and chondro/osteogenic markers by FOP iMPCs is not convincing and more specific chondro/osteogenic markers should be analyzed.

We edited the section on scRNAseq results to better clarify our results and conclusions. Although at first glance it may seem that the muscle stem cells are too diluted in the dataset, we note that despite the overall number of cells per sample (7309 cells after filtering), the proportion of the population of interest (1009 cells or 13.8%) is still higher than physiological proportion (2-7%, Yablonka-Reuveni et al., 2011 J Histochem Cytochem). While higher cell numbers are often preferable, our studies using 1000 muscle cells is in fact a reasonable number: Past studies on rare population have been performed on similar or lower cell number (Rubenstein et al., 2020 Scientific reports; De Micheli et al., 2020 Cell Reports). In addition, using the Seurat algorithm (https://satijalab.org/howmanycells/) we would need only 624 cells to be 95% confident that our sample contains at least 5 cells from each cell types. Thus, we believe our analyses are reflective of the underlying biology we are observing.

We analyzed the expression of typical chondro/osteogenic markers that have been found to be altered in other cell types carrying the *ACVR1*^*R206H*^ mutation. This is now presented in Figure 5—figure supplement 1F,G. *SP7* (Osterix) was not found to be expressed. Overall, those markers have very low level of expression, relatively few cells expressed them and none of them were significantly differentially expressed. *Sox9* expression is increased in FOP clusters.

We also analyzed the expression of cinq genes (*CYR61*, *CTGF*, *NOV* and *WISPs*) which have been shown to be induced during the process of trauma-induced HO and where WISP1 may have act as a negative regulator (Hsu et al., 2020 JCI Insight). None of those genes were significantly differentially expresse; however, CTGF expression was increased in FOP cells. Because this does not significantly impact our conclusions, these data are provided for the reviewer but not included in the current version of the manuscript.

16. Fewer cells were analysed in the control sample compared to FOP sample. The authors should indicate whether analyses were performed to compensate for differences in the sequencing depth.

As noted by the reviewer, the reads per cell value differs between the control and FOP samples. However, as indicated in Figure 4-Source Data1, the Median Genes per cells is equivalent, thus compensating for the reads/cell difference and making our comparison relevant. Hence, we did not perform compensatory analyses, but have chosen to represent our results using Violin plots. These plots enable visualization of the probability density of a given gene expression in various samples, thus minimizing cell number bias between samples.

17. In Figure 4, very few Pax7+ and Myod + cells are detected in the myogenic cluster. To what extent are the differences between control and FOP samples due to non pax7+ cells?

We thank the reviewer for raising this important question. One major limitation of scRNAseq is the low sensitivity that doesn’t detect all transcripts; therefore, scRNAseq analyses use unsupervised clustering as an unbiased way to identify various cell populations and subtypes. At the time this study was performed, the 3’ v2 kit from 10X Genomics was standard, and thus was used for this study. Additional published studies from our groups and others on human satellite cells have shown that this newer version can detect higher proportion of PAX7 expressing satellite cells (Barruet et al., 2020 *eLife*; Dell’Orso et al., 2019 Development). Despite the low amount of non-PAX7+ cells we were still able to confirm findings from our FOP primary Hu-MuSCs showing an increase in ECM and chondro-osteogenic markers.

18. The non-myogenic clusters should be better characterized in Figure 4. The myogenic clusters should also be better defined in Figure 5, especially clusters 0 and 2.

Thank you for this suggestion. The non-myogenic clusters have been characterized and properly labelled in Figure 4 and Figure 5.

19. In Figure 5, cluster # should be specified in panel B.

Cluster numbers are now specified in 5A (see reviewer’s comment #18).

20. The pseudotime analyses are were very confusing to the reviewers and the suitability and rigor of this analysis was questioned. The starting point A contains cluster 4 cells that are MYOG+ and MYOD1+. In panel D, how can differentiation markers exhibit a non-linear progression? Chondro/osteogenic markers should be validated and more specific.

We re-performed the pseudotime analyses in Figure 5 D-F as suggested, and adjusted the branch order for the pseudotime trajectories of myogenic markers for more clarity. While pseudotime analysis does not always reflect linear time, it can be useful to map gene expression changes that occur between cell populations. HuMuSC differentiation and return to quiescence is a dynamic process. Indeed, under the right stimuli, HuMuSCs undergo asymmetric division after proliferation by either self-renewal to replenish their pool or differentiation to form myotubes (Feige et al., 2018 Cell Stem Cell). It is highly probable that similar changes are happening in our myogenic differentiation thus exhibiting a non-linear progression of the myogenic markers. Indeed, our scRNAseq analysis shows different states of muscle stem cells in Figure 5B. The tip of branch A is now constituted of cycling muscle stem cells that have low expression of *MYOG*. The drop of *MYOD1* expression can be associated with cells returning to a more quiescent state such as cluster 0 which is located along to more proximal end of branch A and in the branches’ intersection (Figure 5F). As expected, the myoblast cluster has higher level of *MYOG* and *MYOD1* expression.

Chrondro/osteogenic markers that have been reported to be upregulated in FOP cells are analyzed and shown in Figure 5—figure supplement1 (see reviewer’s comment #15).